# Non-Coding RNA in Type 2 Diabetes Cardio–Renal Complications and SGLT2 Inhibitor Response

**DOI:** 10.3390/ijms262211198

**Published:** 2025-11-19

**Authors:** Elena Rykova, Elena Shmakova, Igor Damarov, Tatiana Merkulova, Julia Kzhyshkowska

**Affiliations:** 1Institute of Cytology and Genetics, Siberian Branch of Russian Academy of Sciences (IC&G SB RAS), Lavrentjev Prospect 10, 630090 Novosibirsk, Russiadamarovis@bionet.nsc.ru (I.D.);; 2Department of Engineering Problems of Ecology, Novosibirsk State Technical University, 630087 Novosibirsk, Russia; 3Laboratory of Translational Cellular and Molecular Biomedicine, National Research Tomsk State University, 634050 Tomsk, Russia; 4The Laboratory of Molecular Therapy of Cancer, Cancer Research Institute, Tomsk National Research Medical Center, Russian Academy of Sciences, 634050 Tomsk, Russia; 5Institute of Transfusion Medicine and Immunology, Institute for Innate Immunoscience (MI3), Medical Faculty Mannheim, University of Heidelberg, 68167 Mannheim, Germany

**Keywords:** type 2 diabetes mellitus, SGLT2 inhibitors, chronic inflammation, diabetic cardiovascular complications, diabetic nephropathy, non-coding RNAs (ncRNAs), signaling pathways, cardiomyocytes, cardiac fibroblasts, endothelial cells, innate immune cells, macrophages, podocytes, tubular epithelial cells, mesangial renal cells, ncRNA genetic variants

## Abstract

Type 2 diabetes mellitus (T2DM) is characterized by an uncontrolled increase in blood glucose levels and insulin resistance in cells of various tissues. Vascular complications in T2DM have an inflammatory nature. Drugs with different mechanisms of action have been developed and used to treat T2DM, initially aimed at controlling blood glucose levels. Among them, sodium-glucose cotransporter 2 inhibitors (SGLT2-i) were developed as specific inhibitors of glucose reabsorption in the kidneys, but along with lowering blood glucose levels, they demonstrated multiple (including non-glycemic) positive effects in the treatment of T2DM related to their beneficial effects on the immune system. SGLT2 inhibitors can reduce the risk of diabetic cardiomyopathy (DCM) and chronic kidney disease (CKD) development in patients with and without diabetes. SGLT2-is improve cardio-renal complications through a number of signaling pathways, including those dependent on the involvement of non-coding RNAs (ncRNAs) and their targets. The best-studied classes of ncRNAs are microRNAs, which are short (less than 200 bases) RNAs (miRNAs), long non-coding RNAs (lncRNAs) (more than 200 bases), and circular RNAs (circRNAs). The regulatory effect of ncRNAs has broad physiological significance, and changes in the ncRNAs’ expression are associated with the pathogenesis of different diseases, including T2DM. RNA-seq allows the construction of networks of interactions of lncRNA/circRNA-miRNA-mRNA called competitive endogenous RNA (ceRNA) networks, to identify clinically significant molecular markers, to improve the mechanistic understanding of pathogenesis, and to contribute to the development of new diagnostics and therapies. Our review summarizes the role of non-coding RNA in the action of SGLT2 inhibitors in cardio-renal complications in T2DM. We focus on methods of detection, genetics, and the effects of non-coding RNA. Specific attention is given to the role of non-coding RNAs in the inflammatory reactions of innate immune cells in relation to the SGLT2 inhibitors.

## 1. Introduction

Diabetes mellitus is a group of metabolic diseases characterized by an uncontrolled increase in blood glucose levels. Type 2 diabetes mellitus (T2DM) accounts for 90–95% of all cases of diabetes mellitus [1]. T2DM is characterized by insulin resistance in cells of various tissues, as well as dysfunction of the beta cells in the pancreas islets [2]. Type 2 diabetes is one of the leading causes of death and can have severe complications, such as blindness, renal and/or heart failure, strokes, and lower limb amputations [1,3,4,5]. Drugs with different mechanisms of action have been developed and used to treat T2DM, which are initially aimed at controlling blood glucose levels. Metformin is currently used as a first-line treatment, which leads to the desired glucose-lowering effect for many patients with T2DM. However, in 30% of cases of patients with T2DM, metformin is ineffective, so other therapeutic agents have been developed and are used.

Currently, among the many drugs used to treat T2DM, sodium-glucose cotransporter 2 inhibitors (SGLT2-i) attract significant attention. These drugs were developed as specific inhibitors of glucose reabsorption in the kidneys, but along with lowering blood glucose levels, they have shown multiple (including non-glycemic) positive effects in the treatment of T2DM. In particular, it turned out that they downregulate the inflammatory processes and, as a result, reduce the risk of developing micro- and macrovascular complications of diabetes [6,7]. Due to the pleiotropic therapeutic effect of these drugs, they are used to treat not only T2DM but also many diseases that occur as a complication of T2DM [8,9]. Clinical studies have shown that SGLT2 inhibitors can reduce the risk of diabetic cardiomyopathy (DCM) and chronic kidney disease (CKD) development in patients with and without diabetes [10,11,12]. Studies of the molecular mechanisms have shown that the SGLT2 inhibitors improve cardio-renal complications through a number of signaling pathways, including those dependent on the involvement of non-coding RNAs (ncRNAs) and their targets.

Increasingly, functional studies on healthy states and different pathologies have shown that ncRNAs are molecules that do not have a traditional protein-coding function. The best-studied classes of ncRNAs that can influence gene expression are microRNAs, which are short (approximately 22 bases) RNAs (miRNAs), long non-coding RNAs (lncRNAs) (more than 200 bases), and circular RNAs [13,14,15]. The regulatory effect of ncRNAs has broad physiological significance, as they are involved in the processes of cell growth and differentiation, organ and tissue development, apoptosis, inflammation, glucose metabolism, intracellular signaling, etc. [14]. Changes in the expression of non-coding RNAs are associated with the pathogenesis of such significant diseases as cancer, cardiovascular diseases, and diseases associated with metabolic disorders, including type 2 diabetes mellitus [16,17,18,19,20,21]. Whole transcriptome sequencing (RNA-seq) provides information on all transcriptional products (ncRNAs and mRNAs) in selected cells and tissues under conditions of interest [22]. RNA-seq results are successfully used to identify non-coding RNAs and construct networks of interactions of lncRNA/circRNA-miRNA-mRNA (competitive endogenous RNA (ceRNA) networks) associated with the development of T2DM and their response to drug therapies, which allows deeper insight into pathogenic molecular mechanisms and contributes to the development of new approaches to diagnosis and treatment [20,23,24,25,26,27,28].

Despite considerable interest in the action of SGLT2-is, the molecular mechanisms involved in their preventive action against T2DM complications remain not fully understood. A number of studies discovering the role of ncRNAs allowed the identification of several signaling pathways associated with their regulatory effects on various types of cardiac and renal cells during the development of T2DM complications. However, only isolated studies were focused on the clarification of the role of ncRNA in the therapeutic effect of SGLT2-is at the cellular level. In particular, their anti-inflammatory action in the innate immune cells (macrophages) was studied in a very limited way. Genetic studies demonstrated that ncRNA variants are associated with an individual’s predisposition to T2DM and its complications, emphasizing the potential mechanistic role of ncRNAs in the regulation of metabolic pathways. Specific ncRNA SNPs can be valuable prognostic markers in the selection of optimal therapy, individualized for each patient. However, currently, the data are still accumulating, and clinical applications are only emerging. This review aimed to elucidate the state-of-the-art in the field in detail, as well as to identify the gaps in knowledge. We highlight here high-throughput studies, and genetic and functional data on ncRNAs’ regulatory role in the T2DM cardio-renal complications, as well as their involvement in the SGLT2-is action that prevents T2DM complications on the heart and kidneys (Figure 1).

## 2. Regulatory Non-Coding RNAs in T2DM

### 2.1. Identification of T2DM miRNA Biomarkers

To date, most of the studies have been focused on the role of individual candidate ncRNAs in the development of T2DM and its complications. As a result, dozens of ncRNAs were identified with a decreased or increased level in various tissues and cell types, both in model animals and in humans (liver, kidney, and muscle biopsies, pancreatic islets, whole blood, PBMC, and CD14+ monocytes), as reviewed elsewhere [29,30,31,32,33,34,35]. Differentially expressed ncRNA circulating in the blood plasma/serum have been identified and considered as valuable biomarkers for the prognosis of T2DM and its complications, especially in the early clinical stage, as well as potential targets for therapy [30,36,37,38] (Table 1). Large-scale analysis by microarray techniques with RT-PCR validation was used for the massive detection of previously unknown ncRNAs associated with T2DM [39,40,41,42,43,44,45]. Further bioinformatics analysis was applied to identify genes associated with differentially expressed ncRNAs, as well as signaling pathways associated with these genes, potentially involved in the pathogenesis of T2DM. Whole-transcriptome studies (RNA-seq) opened new horizons in the study of the role of ncRNAs in the development of T2DM. This approach allows identification of networks of competing endogenous RNAs (ceRNAs), which are in dynamic interaction, that are crucial for the development of the pathology [20,34,46,47,48,49,50]. This research field is highly dynamic, and rapidly accumulating data bring new knowledge about the role of ncRNA in the pathogenesis of T2DM.

### 2.2. Circulating miRNA as Biomarkers of T2DM

Diabetes is characterized by an insufficient function of pancreatic beta cells, leading to relative or absolute insulin deficiency, as well as to a reduced sensitivity of peripheral target tissues to insulin [33]. Recently, Yang et al. summarized miRNAs that are involved in the beta cell proliferation and function, including insulin synthesis and secretion [33]. In a high-fat diet-induced diabetes mouse model, whole-transcriptome analysis of pancreatic islets identified novel regulatory axes associated with impaired insulin secretion (miR-6948-5p/Cacna1c, miR-6964-3p/Cacna1b, miR-3572-5p/Hk2, miR-3572-5p/Cckar, and miR-677-5p/Camk2d) and increased pancreatic beta-islet mass (miR-216a-3p/FKBP5, miR-670-3p/Foxo3, miR-677-5p/RIPK1, and miR-802-3p/Smad2 и ENSMUST00000176781/Caspase9) (Table 1) [34].

Accumulated results allowed them to propose circulating miRNA as potential biomarkers for the pre-diabetic and early diabetes states [37,51,52] and T2DM [23,34,52,53,54], as well as biomarkers that discriminate between five previously defined subtypes that characterize the differences in pathophysiological processes in T2DM: severe insulin-resistant diabetes (SIRD), severe insulin-deficient diabetes (SIDD), mild age-related diabetes (MARD), mild obesity-related diabetes (MOD), and mild early-onset diabetes (MEOD) (Table 1) [55]. Song et al., using RNA-seq, found a number of differentially expressed lncRNAs, miRNAs, and mRNAs in the plasma of newly diagnosed T2DM patients [23]. Analysis using GO, PPI network, and lncRNA-miRNA-mRNA network revealed that the differentially expressed mRNAs of SLC25A4, PLCB1, AGTR2, PRKN, and SCD5 genes might be under the control of miR-199b-5p, miR-202-5p, miR-548o-3p, and miR-1255b-5p [23]. The discovered miRNAs potentially associated with T2DM pathogenesis identified in large-scale studies represent potential markers and targets for therapy, which requires further experimental verification (Table 1).

**Table 1 ijms-26-11198-t001:** NcRNAs as biomarkers of T2DM.

ncRNA	Expression Level	Comparison Groups	Source	Identification Method	Ref
miR-375, miR-30a, miR-34a, miR-7a, miR-486-5p miR-17-92 cluster, miR-130a-3p, miR-130b-3p, and miR-152-3p	Up	Mouse models of T2DM/healthy controls	Murine pancreatic beta-cells, pancreatic beta-cell lines	qRT-PCR	[33]
miR-6948-5p; miR-6964-3p; miR-677-5p; miR-670-3p; 12_4382	Up	Mice on a high-fat diet/low-fat diet	Pancreatic islets	RNA-seq	[34]
miR-3572-5p; miR-216a-3p; miR-802-3p; miR-1188-5p	Down				
hsa-miR-4660; hsa-miR-451a; hsa-miR-3146	Up	Patients with SIRD/healthy individuals	Blood serum	RNA-seq	[55]
hsa-miR-221-5p; hsa-miR-6852-5p; hsa-miR-224-5p; hsa-miR-199a-5p; hsa-miR-30e-3p; hsa-miR-1301-3p; hsa-miR-214-3p; hsa-miR-744-5p; hsa-miR-766-3p; hsa-miR-1307-3p; hsa-miR-432-5p	Down				
hsa-miR-3143; hsa-miR-942-3p; hsa-miR-20b-5p; hsa-miR-576-5p; hsa-miR-548ay-5p; hsa-miR-548d-5p; hsa-miR-454-5p; hsa-miR-324-3p; hsa-miR-1843	Down	Patients with SIDD/healthy individuals			
hsa-miR-548s	Up	Patients with MARD/healthy individuals			
hsa-miR-3928-3p; hsa-miR-378c	Down				
hsa-miR-129-5p; hsa-miR-548bc; hsa-miR-3614-5p; hsa-miR-6866-5p; hsa-miR-6741-5p; hsa-miR-320a-3p; hsa-miR-5000-3p; hsa-miR-320e; hsa-miR-576-3p	Up	Patients with MOD/healthy individuals			
hsa-miR-6837-3p; hsa-miR-6763-5p; hsa-miR-144-5p; hsa-miR-625-5p; hsa-miR-30e-3p; hsa-miR-628-3p; hsa-miR-152-3p; hsa-miR-570-3p; hsa-miR-584-5p; hsa-miR-26b-5p; hsa-miR-1197; hsa-miR-3177-3p; hsa-miR-659-5p; hsa-miR-1271-5p; hsa-miR-361-5p; hsa-miR-628-5p; hsa-miR-181a-5p; hsa-miR-191-5p	Up	Patients with MEOD/healthy individuals			
hsa-miR-486-5p	Down				
miR-30a-5p	Up	Patients with T2DM/healthy individuals	Whole blood	qRT-PCR	[54]
miR-30d	Up	Patients with T2DM/healthy individuals	Blood serum		
miR-34a; miR-146a	Up	Patients with T2DM/healthy individuals	PBMCs		
miR-320a; miR-126; miR-21; miR-15a; miR-145	Down	Patients with T2DM/healthy individuals	Blood plasma		
miR-223	Down	Patients with prediabetes/healthy individuals	Serum microvesicles		
miR-150	Up	Rats with T2DM/rats without T2DM	Cardiomyocytes		
miR-103	Down	Rats with T2DM/rats without T2DM	PBMCs		
miR-126-3p; miR-223-3p; miR-21-5p; miR-15a-5p; miR-24-3p	Down	Patients with T2DM/healthy individuals	Blood plasma	qRT-PCR	[53]
miR-34a-5p; miR-148a-3p; miR-30d-5p	Up				
miR-199b-5p; miR-548o-3p	Up	Patients with T2DM/healthy individuals	Blood plasma	RNA-seq	[23]
miR-202-5p; miR-1255b-5p	Down				
ENST00000381108.3; ENST00000515544.1; ENST00000539543.1; ENST00000508174.1; ENST00000564527.1	Up	Patients with T2DM/healthy individuals	PBMCs	Microarray	[42]
TCONS_00017539; ENST00000430816.1; ENST00000533203.1; ENST00000609522.1; ENST00000417079.1	Down				
XR_108954.2	Up	Patients with T2DM/healthy individuals	PBMCs	Microarray+ qRT-PCR	
RP4-605O3.4; AC074117.2	Down	Patients with T2DM or prediabetes/healthy individuals	Blood serum	qRT-PCR	[51]
MALAT1; TUG1; MIAT; NEAT1	Up	Patients with T2DM/healthy individuals	Blood serum	qRT-PCR	[52]
circRNA_008565	Down	Rats with T2DM/rats without T2DM	Pancreatic β-cells	Microarray+ qRT-PCR	[33]
circRNA_0054633	Up	Women with gestational diabetes mellitus/healthy women	Blood serum	qRT-PCR	

### 2.3. lncRNA and circRNA as Biomarkers of T2DM

lncRNAs and circRNAs have been intensively studied during the last several years in the context of T2DM pathogenesis (Table 1). lncRNA MALAT1, lncRNA steroid receptor RNA activator (SRA), and lncRNA βFaar are involved in beta cell proliferation and function, while lncRNA MEG3 and lncRNA HOTAIR are involved in insulin resistance [33]. To identify new lncRNAs as potential molecular markers of T2DM, Lin et al. [40] used bioinformatics analysis of available transcriptome data and constructed a T2DM-associated ceRNA network consisting of 98 genes, 86 microRNAs, and 167 lncRNAs. The highest degree nodes were VEGFA/ESR1, hsa-miR-21, and lncRNA MIR22HG, respectively [40]. In the study of Ma Q et al. [42], 68 differentially expressed lncRNAs and 74 differentially expressed mRNAs were identified in PBMCs from patients with T2DM compared with healthy donors, their function being in the multiple biological processes associated with diabetes [42]. Ali et al. aimed to more accurately determine those at risk of developing prediabetes and progressing to T2DM by combining several preselected circulating RNA biomarkers [51]. This study identified a novel STING/NOD/IR RNA panel (TMEM173, CHUK mRNAs, miR-611, and miR-1976 and lncRNA RP4-605O3.4) as a pre-DM and T2DM-associated biomarker panel [51]. Su et al. performed a systematic review and meta-analysis of human case–control or cohort studies on differential lncRNA expression in T2DM that allowed them to conclude that lncRNAs may be promising diagnostic markers for prediabetes, T2DM, and its complications [52].

circRNA CDR1as has been reported to modulate insulin production and transcription in islet cells by inhibiting miR-7 through regulating endogenous target genes Pax6 and Myrip [56]. Circ-Tulp4, circANKRD36, and circRNA_008565 are examples of circRNAs important for the beta cells’ function and insulin secretion. Insulin sensitivity was found to be regulated in the peripheral tissues by circRNA_0054633, circH19, and circRNA_000203 [33] (Table 1).

## 3. Regulatory Non-Coding RNAs in Diabetic Cardiovascular Complications

### 3.1. Cardiovascular Complications in T2DM

Emerging studies have shown that the risk of adverse cardiovascular events increases several times for T2DM patients compared with clinically healthy individuals [35,50,57]. T2DM can lead to the development of diabetic cardiomyopathy (DCM), heart failure (HF), and myocardial infarction (MI) [35,50]. Epidemiological studies have confirmed a correlation between T2DM markers, such as elevated serum glucose, glycated hemoglobin levels, impaired glucose tolerance, and increased risk of heart diseases [58,59,60,61]. Continuous cascade reactions induced by hyperglycemia and insulin resistance lead to chronic inflammation in T2DM, where innate immunity has a central role [62,63,64,65]. The inflammatory factors produced by monocytes and macrophages in metabolic conditions induce pathological activation of endothelial cells, as well as apoptosis and pyroptosis of cardiomyocytes, which aggravate the progression of DCM [66,67,68,69,70]. DCM can be clinically manifested as a restrictive phenotype, which eventually evolves into HF with preserved ejection fraction (HFpEF) and a dilated phenotype, which evolves into HF with reduced ejection fraction (HFrEF) [71]. DCM is a unique cardiomyopathy regarding the molecular mechanism of its development related to the metabolic dysregulation in the heart [35]. DCM is characterized by changes in the myocardial tissue induced by impaired insulin signaling via dysregulated adenosine monophosphate-activated protein kinase (AMPK), protein kinase C (PKC), and mitogen-activated protein kinase (MAPK) signaling, leading to fibrosis by increased collagen deposition and the changed protein structure in the extracellular matrix. Fibrotic process is also enhanced by the epithelial–mesenchymal transition (EMT) under hyperglycemic conditions, as well as abnormal lipid metabolism, with increased fatty acid uptake leading to the development of lipotoxicity, endothelial dysfunction, and atherosclerosis.

### 3.2. miRNAs in Diabetic Cardiovascular Complications

An increasing number of studies confirm the role of miRNAs in the regulation of pathophysiological alterations associated with diabetic heart injury [30,35,57,72]. Clinical studies highlight the significance of certain miRNAs as potential diagnostic biomarkers for T2DM-associated cardiovascular complications. A recent review by Kura et al. [35] summarizes the differentially expressed miRNAs in DCM that are associated with oxidative stress (miRNA-21, miRNA-141, miRNA-1, miRNA-200c, miRNA-503, miRNA-22, miRNA-133a/b, miRNA-144, miRNA-499), inflammation (miRNA-21, miRNA-92a, miRNA-204, miRNA-24, miRNA-146a), cardiac hypertrophy (miRNA-195, miRNA-208a, miRNA-451, miRNA-133a, miRNA-150, miRNA-30c), myocardial fibrosis (miRNA-21, miRNA-122-5p, miRNA-150-5p, miRNA-9, miRNA-503, miRNA-142, miRNA-133a, miRNA-700, miRNA-495), apoptosis (miRNA-1, miRNA-34a, miRNA-195, miRNA-208, miRNA-320, miRNA-21, miRNA-181a, miRNA-30c), and mitochondrial dysfunction and metabolic disturbances (let-7, miRNA-141, miRNA-223, miRNA-320, miRNA-30c) [35].

A number of miRNAs have been identified as highly expressed in cardiomyocytes, playing significant roles in the healthy state of the heart and in cardiac pathology, among them miRNA-1, miRNA-16, miRNA-27b, miRNA-30d, miRNA-126, miRNA-133, miRNA-143, miRNA-208, and the let-7 family [35,73,74,75] (Figure 2).

For example, abundantly expressed in normal cardiac tissue, miR-133a was significantly downregulated in DCM mice. An in vitro study showed that miR-133a attenuated cardiomyocyte hypertrophy induced by hyperglycemia [76]. miRNAs, such as miR-186-5p, miR-22, and miR-29a, are shown to suppress hyperglycemia-induced apoptosis in cardiomyocytes in DCM, thereby potentially offering therapeutic targets to protect cardiomyocytes [77]. In contrast, miRNAs like miR-34a, miR-483, and miR-207 promote apoptosis and exacerbate the loss of cardiomyocytes [77]. Notably, miR-144 exhibits a dual role by targeting different mRNAs, thereby both promoting and suppressing apoptosis depending on the specific context within the hyperglycemic environments [77].

Endothelial cells in the myocardium modulate vascular tone and structure. Certain endothelial-enriched miRNAs have been found to have effects on cardiac function and remodeling, such as miR-126, miR-92a, miR-155, miR-222, and miR-221 [30,35]. EMT plays a significant role in the DCM pathogenesis. A number of microRNAs were found to be upregulated in cardiac endothelial cells in DCM, such as miR-21, miR-92a miR-200c, and miR-195-5p miRNA-204, while miR-18a-5p miR-142-3p, miR-222, miR146a, miRNA-24, miR-9, and miR-126 were downregulated [30,35,74,78,79] (Figure 2). An interesting circulating prognostic marker is miR-92a, which increases several years before the development of coronary artery diseases in T2DM [54]. Cardiac fibroblasts are highly abundant cells in the myocardium and are essential for maintaining the heart’s extracellular matrix. In response to stress or injury, they rapidly proliferate and produce excess extracellular matrix proteins, leading to cardiac fibrosis. A number of differentially expressed miRNAs have been shown to be associated with pathological processes in cardiac fibroblasts, among them upregulated miR-1, miR-21, miR-24, miR-122, and miR-150-5p, and downregulated miR-135b, miRNA-141, and miR-495 [30,35,74,79,80] (Figure 2). The involvement of miR-141 in the development of diabetic cardiomyopathy was shown in the work of Che H, et al., who found a decrease in miR-141 levels in a streptozotocin (STZ)-induced mouse T2DM model and in the cultured mouse neonatal cardiac fibroblasts under the influence of high glucose levels [81]. The authors found in primary cardiac fibroblasts under HG conditions an increased level of lncRNA MALAT1, which is a sponge for miR-141. The work demonstrated the central role of miR-141 by reducing the expression of *NLRP3* and *TGF-β1* target genes in the anti-fibrotic effect of the hormone melatonin, since suppression of miR-141 expression abolished this effect in cultured cardiac fibroblasts [81].

It is of note that within 5 years of diabetic cardiomyopathy onset, increasing cardiac hypertrophy assessed using cardiac magnetic resonance (CMR) was independent of glycemic control and paralleled by the upregulation of the circulating miR122-5p targeting the extracellular matrix *MMP2* gene expression [80]. Jakubik et al. summarized published data and generated the network of ncRNAs and their target genes involved in the pathophysiology of DCM [30]. Cardiac fibrosis, hypertrophy, oxidative stress, inflammation, apoptosis, autophagy, and pyroptosis were presented. miR-146a and miR-195 appeared to be the most promising miRNAs as regulators in DCM, since they can target at least six different genes and can regulate different biological processes involved in DCM [30].

As a result of high-throughput miRNA studies, the list of potential circulating miRNA markers for DCM has been increased. Bielska A et al. conducted a miRNA expression profiling study and validated the diagnostic significance of miR-615-3p, miR-3147, miR-1224-5p, miR-5196-3p, miR-6732-3p, and miR-548b-3p for DCM. All tested miRNAs showed high diagnostic value. Notably, the miRNAs tested under the study were more effective than the non-specific inflammatory parameters like chemokines CXCL12 and macrophage migration inhibitory factor (MIF) [44]. A unique profile of plasma-derived miRNAs in T2DM patients with CAD was revealed in the recent study of Szydełko et al. [45]. The authors used expression profiling of 2578 miRNAs by microarray techniques that allowed them to develop a panel of hsa-miR-4505, hsa-miR-4743-5p, and hsa-miR-4750-3p. The authors postulate that this panel could serve as a novel non-invasive biomarker for T2DM-CAD patients compared with T2DM subjects and controls [45]. For coronary artery diseases (CAD) that develop against the background of T2DM, a panel of circulating microRNAs has been identified; the differential expression levels of which may be diagnostically significant [54]. The level of some miRNAs increases in coronary artery disease, for example, miR-1, miR-132, miR-133, and miR-373-3-p [54]. In a recent study, Mansuori et al. showed elevated plasma miR-1 and miR-133a levels in three groups of patients with myocardial infarctions (MI): MI patients, MI patients with pre-diabetes (metformin non-users), and MI patients with diabetes (metformin users) compared with healthy donors [82]. Also, miR-1 and miR-133a levels were lower in metformin-user patients than in non-user patients, indicating a potential mechanism for cardiac protection by the treatment [82].

Recently, Yin et al. published a systematic review of the contribution of circulating miRNAs as biomarkers of diabetic complications, based on analysis of the selected 71 studies screened and 75 miRNAs validated [38]. Using KEGG pathway enrichment analysis, they demonstrated that in DCM, miR-130, miR-21, and miR-199 are implicated in a considerable number of pathways, among them the fatty acid synthesis and metabolism pathway was significantly enriched [38].

### 3.3. lncRNA and circRNA in Cardiovascular Complications

Accumulating data demonstrate that lncRNAs may participate in the development of cardiac hypertrophy and heart failure in DCM through the regulation of oxidative stress, inflammation, cardiac hypertrophy, myocardial fibrosis, apoptosis, mitochondrial dysfunction, and metabolic disturbances [28,59,77,83,84]. lncRNAs associated with diabetic cardiomyopathy, according to the rodent DCM models and cultured cells studies, include DCRF, DACH1, TINCR, Kcnq1ot1, ZNF593-AS, MALAT1, Airn ZFAS1, H19, GAS5, HOTAIR, Crnde, NEAT1, MIAT, NORAD, AK081284, ANRIL, NONRATT007560.2, MEG3, and PVT1 (as reviewed by [59]).

Meng et al. found that lncRNA TINCR was significantly up-regulated in an STZ-induced DCM rat model, promoted cardiomyocyte pyroptosis, and aggravated cardiac dysfunction [85]. They further found that TINCR interacted with NLRP3 and stabilized NLRP3 mRNA, thereby accelerating the initiation and progression of DCM. NON-RATT007560.2 [86], HOTAIR [87], and ANRIL [88] are associated with cardiac remodeling in DCM through the regulation of cardiomyocyte apoptosis and oxidative stress [84]. Another lncRNA, AK081284, was proven to be associated with cardiac interstitial fibrosis via TGF-β1. Zhang et al. discovered that AK081284 mediated the effect of IL-17 on interstitial fibrosis in the diabetic heart [89]. In addition, the lncRNA Crnde, which was primarily increased in CFs with TGF-β stimulation, was found to exert a protective effect against cardiac fibrosis [90].

The lncRNA-miRNA-mRNA axis, which is involved in the pathophysiological processes of cardiomyocytes, plays a significant role in instigating and advancing the development of DCM [77]. Wu et al. emphasized the advancements related to the lncRNA-miRNA-mRNA axis concerning cardiac apoptosis in the context of DCM progression. They reviewed the data demonstrating the involvement of MALAT1/miR-181a-5p/miR-22 axis, GAS5/miR-138/miR-320/miR-126a/b-5p axis, H19/miR-675 axis, KCNQ1OT1/miR-181a-5p axis, MIAT/miR-22-3p axis, and AK139328/miR-204-3p axis [77]. Each lncRNA is involved in a number of lncRNA-miRNA-mRNA axes taking part in diabetic heart disease progression. As an example, lncRNA KCNQ1OT1 expression was found to be increased in T2DM patients, high glucose-induced cardiomyocyte model, and diabetic mouse models. KCNQ1OT1 knockdown suppresses pyroptosis in high-glucose-induced AC16 cells and primary cardiomyocytes by affecting miR-214-3p and Caspase-1 levels, and also restores cytoskeletal structure and reduces calcium overload in cultivated cardiomyocytes, improving cardiac structure and function [91]. According to Zhao et al., KCNQ1OT1 and its target gene miR-181a-5p control cardiomyocyte apoptosis in DCM by influencing the regulation of the *PDCD4* gene [92]. Recently, Leng et al. reported that KCNQ1OT1/circ_0020316-miR-92a-2-5p-MAPK3 regulatory networks were involved in vascular injury in T2DM [93].

High-throughput techniques application allows for the discovery of novel ncRNAs, which were not known before to be associated with diabetes pathogenesis. Microarray data from the Gene Expression Omnibus (GEO) (GSE26887 dataset) were used to construct a ceRNA network associated with diabetic cardiomyopathy [41]. The results revealed that the lncRNA XIST and its two interacting miR-424-5p and miR-497-5p were involved in the pathogenesis of cardiomyopathic complications in T2DM patients [41]. Recently, a high-fat diet (HFD) mouse model was used to address the mechanisms that link HFD-induced cardiac injury and myocardial infarction [50]. High-throughput sequencing identified differentially expressed mRNAs and long non-coding RNAs in diabetic mice with MI [50]. Heart tissue lncRNA and mRNA expression profiles were obtained, which allowed the construction of an lncRNA-gene network in the HFD-treated MI group. Three core lncRNAs, AC165273.2, 2310039LI5Rik, and AC160401.1, were identified in the network map and selected for further analysis. Analysis of the shared intersections among these core lncRNAs identified eight target genes; among them, *Rapgef5* and *Ing1* were involved in proliferation and apoptosis of cardiomyocytes induced by HG/HL with hypoxia/reoxygenation [50].

Recent studies have confirmed that differentially expressed circRNAs have a vital role in diabetic complications [83,94]. Li et al. aimed to identify the expression characteristics of circRNAs in the peripheral blood of CAD patients and T2DM patients [95]. Using microarray analysis of circRNA in RNA samples from whole peripheral blood, followed by a verification study on the independent cohort, they identified hsa-circRNA11783-2 as significantly related to CAD and T2DM [95]. Several circular RNAs have been implicated in DCM progression, including circ_000203, circRNA_010567, circHIPK3, CACR, and circCDR1as, all of which were upregulated in DCM development [83,94,96]. circRNA_010567, circHIPK3, and circRNA_000203 are upregulated in the heart tissues of diabetic mice and are related to the myocardial fibrosis pathogenesis [97,98,99]. Hsa_circ_0076631, called caspase-1-associated circRNA (CACR) as well, has elevated expression in the hyperglycemia-exposed cardiomyocytes and in the patients with diabetes [100]. CircCDR1as was upregulated in DCM hearts of STZ-induced diabetic mice, promoting cardiomyocyte apoptosis [83,94]. Recently, Yuan et al. has discovered that circRNA mm9_circ_008009 is downregulated in mouse hearts with DCM and in cardiomyocytes treated with the advanced glycation end products (AGEs). They named it, as well as its conserved human circular RNA hsa_circ_0131202, DICAR (the diabetes-induced circulation-associated circular RNA). DICAR was able to efficiently inhibit the HG-induced pyroptosis in the primary mouse cardiomyocytes and in the HL-1 and HCM cardiomyocyte cell lines via the interaction with valosin-containing protein (VCP) and blocking of the Med12 protein degradation [101].

Accumulating data support the potential of several ncRNAs as valuable markers of T2DM-associated cardiac disorders. Dysregulated ncRNAs have been proven to be involved in the DCM pathogenesis, when studied in patients, as well as in animal models in vivo and cell cultures in vitro. However, the reviewed studies demonstrate sometimes contradictory results due to different human heart pathologies and corresponding animal models under investigation. It is obvious that further collaborative laboratory and clinical studies are urgently needed to enforce progression in the field.

## 4. Regulatory Non-Coding RNAs in Diabetic Nephropathy

### 4.1. Renal Complications in T2DM

Normal renal function is the result of cooperative work of a number of highly specialized epithelial, endothelial, immune, and interstitial cell types and subtypes [102]. Diabetic nephropathy (DN) is a severe microvascular complication of diabetes, characterized by renal interstitial inflammation and fibrosis. It is developed due to the hyperglycemic (HG) conditions in T2DM, which promotes the formation of glycosylation end products that activate signaling pathways, including protein kinase C (PKC), transforming growth factor-β (TGF-β), mitogen-activated protein kinase (MAPK), stress-activated protein kinase/jun N-terminal kinase (SAPK/JNK), Janus kinase/signal transducer and activator of transcription (JAK/STAT3), nuclear factor-κB (NF-κB), PI3K/AKT, Nrf2/ARE, and AMPK [103,104]. These pathological events in the renal cells result in downstream effects, including impaired antioxidant defense (via inhibition of Nrf2), enhanced inflammation (through NF-κB and JAK/STAT3), fibrotic signaling (via TGF-β1 and MAPK components p38, JNK, and ERK), and inhibition of autophagy (through suppressed AMPK and altered mTOR signaling) [103,104].

HG conditions activate the renin-angiotensin-aldosterone system (RAAS) that accelerates renal damage by increasing calcium influx into podocytes, stimulating proinflammatory cytokines, matrix metalloproteinase-9 (MMP-9), transforming growth factor-β (TGF-β), and activating macrophages. This also enhances aldosterone secretion, upregulating profibrotic factors and promoting kidney fibrosis [104,105]. DN manifests histologically as a thickening of the glomerular and thylakoid membranes, resulting in podocyte loss, leading to glomerulosclerosis, renal interstitial inflammation, fibrosis, and progressive renal function decline [103].

### 4.2. miRNAs in Diabetic Nephropathy

In both human clinical and animal experimental studies, the altered expression of regulatory RNAs, such as miRNAs, lncRNAs, and circRNAs, was reported in DN in clinical samples, including serum, plasma, whole blood, and tissues [38,106,107,108,109]. Abdelmaksoud et al. comprehensively reviewed the role of miRNAs in DN, specifically focusing on their impact on key signaling pathways implicated in DN progression [106]. They summarized the results coming from the primary and cultured renal cells and experiments in animal DN models, as well as clinical trials, indicating that miRNAs regulate several essential signaling pathways, including Notch, JAK/STAT, Wnt/β-Catenin, PTEN/PI3K/AKT/mTORC1, and TGF-β-mediated pathways. miR-145–5p, miR-30 family (miR-30a/b/c/d/e), and miR-146a downregulated the Notch signaling pathway in human podocytes and tubular epithelial cells, thus preventing fibrosis, podocyte damage, and inflammation in DN [106,110,111,112]. In contrast, miR-135a activates the Notch pathway in renal cells; therefore, inhibiting miR-135a causes diabetic rats to have less kidney fibrosis [113]. miR-150, miR-214, miR-155, and miR-33-5p upregulate the JAK/STAT pathway and were shown to be increased in DN, thus inducing its progression [114,115,116,117]. miR-20a and miR-30c-5p expression levels were found to be decreased in the renal tissues of patients with diabetic kidney disease; therefore, their overexpression was proposed to provide a potential therapeutic effect in DN by JAK/STAT suppression [118,119]. The aberrant activation of the Wnt/β-catenin pathway results in podocytes and mesangial renal cell damage, leading to the enhancement of diabetic kidney disease. miR-27a, miR-21, miR-671–5p, miR-135a/b, miR-221, miR-17–5p, miR-466o–3p, and miR-499–5p were shown to play roles in DN models in vivo and in vitro by regulation of the Wnt/β-catenin pathway [120,121,122,123,124,125]. miR-214, miR-26a, miR-22, miR-141–3p, miR-21, miR-188–5p, miR-142–5p, miR-155–5p, and miR-181b-5p have been found to be upregulated in different models of DN, inducing PI3K/PTEN/AKT/mTOR signaling pathway activation predominantly by suppressing PTEN expression [39,106,126,127,128]. The TGF-β pathway is strongly linked to DN, as far as its activation is associated with renal hypertrophy, inflammation, fibrosis, and glomerular epithelial cell injury. Aberrant expression of miR-21, miR-155, miR-146a, miR-92b-3p, miR-27a, miR-135a-5p, miR-192, miR-200a, miR-10a/b, miR-7a-5p, and miR-26a leads to renal complications by abnormal regulation of the TGF-β pathway in DN [106,129,130,131,132,133,134]. miRNA expression changes in renal cells are summarized in Figure 2.

NcRNA-mediated autophagy is involved in various pathological processes of DN, such as cell apoptosis, inflammation, mitochondrial dysfunction, renal injury, and fibrosis. miRNA and lncRNA affect autophagy during DN progression [107]. miR-217, miR-155, miR-34a, miR-150-5p, miR-1187, miR-7002-5p, miR-218, miR-379, miR-214, miR-22, miR-192, miR-32, and miR-543 are listed as autophagy-inhibiting miRNAs in podocytes and tubular epithelial cells, leading to renal damage in DN [107]. miR-451 was reported to be downregulated in HG-treated human kidney-2 (HK-2) cells and db/db mice, contributing to renal injury and fibrosis, as well as increased expression of miR-451 was shown to alleviate these effects [135].

Biomarkers of the early DN state and predictors of the diabetic kidney disease development risk are of importance for the personalized T2DM treatment. Bijkerk R et al. were the first to conduct pre-screening of circulating miRNA profiles in three healthy donors and eight DN patients, followed by a larger study confirming that DN patients had lower circulating levels of 11 miRNAs (miR-25, miR-27a, miR-126, miR-130b, miR-132, miR-152, miR-181a, miR-320, miR-326, miR-340, and miR-660) compared to healthy controls [136].

Recently, Yin et al. has published a systematic overview to collect and summarize the current trials of T2DM patient blood miRNAs as promising diagnostic biomarkers for diabetic complications. Authors reviewed a total of 24 differentially expressed miRNAs selected from the literature screened for DN, as follows: hsa-miR-9-3p, miR-15a-3p, miR-16-1-3p, miR-21-3p, miR-25-3p, miR-27a-3p, miR-29c-3p, miR-31-3p, miR-126-3p, miR-130-3p, miR-132-3p, miR-152-3p, miR-192-3p, miR-200a-3p, miR-214-3p, miR-223-3p, miR-320a, miR-326, miR-340-3p, miR-377-3p, miR-378a-3p, miR-451a, miR-574-3p, and miR-660-3p [38]. miR-126 and miR-192 occurred three and four times, respectively; both of these two miRNAs were significantly downregulated in DN patients [38]. According to KEGG pathway enrichment analysis, the top three significantly enriched miRNAs in DN were miR-27, miR-25, and miR-29. Notably, T2DM patients with renal and cardiovascular complications have differentially expressed common miRNAs such as miR-29 and miR-126 [38]. Mechanistically, the involvement of miR-29 in the development of glomerular fibrosis may be related to TGF-β/SMAD signaling pathway regulation [137]. miR-126 can suppress inflammation and ROS production in human umbilical vein endothelial cells (HUVECs) treated with high glucose by modulating the HMGB1 expression [138,139].

Exosomes are cell-generated small extracellular vesicles that carry a number of active biomolecules, including microRNAs, which can influence various pathological processes associated with kidney diseases. Plasma exosomes do not normally pass through the glomerular filtration barrier; therefore, the exosomes present in the urine tend to originate mainly from cells of the urogenital system, reflecting the state of the kidneys [104,140]. There is a large number of discovered exosomal biomarkers that correlate with various stages of DN progression. Urinary exosomal miR-133b-3p, miR-342-3p, and miR-30a-5p were found to be elevated in diabetic patients with normoalbuminuria and increased significantly with the progression to micro- or macro-albuminuria in patients with T2DM, potentially serving as early indicators of DN [104,140]. Exosomal miR-21-5p and miR-23b-3p in the urine were elevated in patients with T2DM and were linked to altered renal function, renal sclerosis, and fibrosis [104], and miR-451 was correlated with renal failure [104,140].

### 4.3. lncRNA and circRNA in Diabetic Nephropathy

Emerging studies discovered lncRNAs involved in DN pathogenesis; however, the functional roles of lncRNAs are still under investigation. There is an increasing body of evidence indicating that lncRNAs can act as competing endogenous RNAs (ceRNAs) and play a critical role in the molecular mechanisms of DN [141]. Functional studies demonstrated that NEAT1/miR-93-5p/CXCL8 and LINC00960/miR-1237-3p/MMP-2 might be potential RNA regulatory pathways to regulate the disease progression of DN [142,143]. Li et al. demonstrated that lncRNA Tug1/PGC1a had a renoprotective effect via regulating mitochondrial remodeling and urea cycle metabolites in diabetic mice [144]. lncRNA Erbb4-IR was reported to promote renal fibrosis via inhibiting miR-29b in db/db mice DN model [145]. lncRNA KCNQ1OT1 affects cell proliferation, apoptosis, and fibrosis through regulating the miR-18b-5p/SORBS2 axis and NF-ĸB pathway in the DN model of HG-treated human glomerular mesangial cells (HGMCs) and human renal glomerular endothelial cells (HRGECs) [146]. The lncRNA SNHG14 level was elevated in the serum of DKD patients and in HG-induced HK-2 cells [147]. Silencing of lncRNA SNHG14 alleviated renal tubular injury via the miR-483-5p/HDAC4 axis in HG-induced HK-2 cells [147]. Studies in the STZ-induced rat DN model and HG-treated HK-2 and 293T showed that lncRNA MALAT1 promotes renal fibrosis and injury in DN [148].

Cao et al. used GEO-based microarray datasets to identify the differentially expressed genes (DEGs) between DN samples and control human tubulointerstitial tissues [43]. Enrichment analyses, construction of protein–protein interaction (PPI) network, and visualization of the co-expressed network between mRNAs and microRNAs (miRNAs) were performed. Finally, they identified *CD53*, *CSF2RB*, and *LAPTM5* as hub genes of tubulointerstitial lesions in DN. Authors speculated that these genes can be closely related to the pathogenesis of DN, and the predicted RNA regulatory pathway of NEAT1/XIST-hsa-miR-155-5p/hsa-miR-486-5p-CSF2RB presents a biomarker axis to the occurrence and development of DN [43]. miR-155 increased levels of miR-155 were detected in the human renal glomerular endothelial cells in DN, leading to the expression reduction in the target gene ETS-1, as well as its targets VCAM-1, MCP-1, and caspase-3, inducing an inflammatory response and apoptosis [149]. HG-induced mouse podocytes also showed the involvement of miR-155; the increased expression of which leads to the suppression of the target gene *BDNF* expression, causing oxidative stress and inflammation in mouse models of DN [150].

A study by Guo et al. intended to construct a ceRNA network in human renal tissue via bioinformatics analysis to determine the potential molecular mechanisms of DN pathogenesis [151]. Based on the analysis of the ceRNA network, five differentially expressed lncRNAs (DE lncRNAs) (SNHG6, KCNMB2-AS1, LINC00520, DANCR, and PCAT6), five DE miRNAs (miR-130b-5p, miR-326, miR-374a-3p, miR-577, and miR-944), and five DE mRNAs (PTPRC, CD53, IRF8, IL10RA, and LAPTM5) were demonstrated to be related to the pathogenesis of DN [151]. Their findings declared a novel underlying molecular mechanism of the SNHG6/miR-944, LINC00520/miR-577, PCAT6/miR-326, DANCR/miR-577, and KCNMB2-AS1/miR-374a-3p signaling axis in promoting DN progression [151]. Recently, the clinical value and function of DANCR in DN were further explored by Kuang et al. [152]. Expression levels of DANCR in the serum of patients with DN or HG-treated human proximal tubular epithelial cells (HK2) were analyzed, demonstrating that an abnormal upregulation of DANCR expression level in DN can induce renal tubular injury via the miR-214-5p/KLF5 axis [152]. In a recent study, Hao et al. established DN models (the STZ-treated mice and HG-stimulated HK-2 cells) in order to evaluate the xanthohumol treatment molecular mechanism [153]. lncRNA SNHG10 was downregulated in the renal tissues of DN mice and HG induced HK-2 cells, and xanthohumol inhibited the progression of DN by regulating SNHG10/miR-378b [153].

A significant component of DN pathogenesis is hyperglycemia-induced ROS generation, leading to the functional disturbance of different types of renal cells, among them renal proximal tubule epithelial cells, podocytes, and mesangial cells being highly important [154]. The involvement of multiple lncRNAs aberrant expression in DN development has been demonstrated for these cell types. Recently, Li et al. reviewed the studies of lncRNAs that modulate processes such as epithelial–mesenchymal transition (EMT), fibrosis, proliferation, and cell death, ultimately contributing to the pathological manifestations of DN [154]. Aberrant regulation of NEAT1/miR-27b-3p/ZEB1 and Dlx6-os1/EZH2/SOX6 pathways in mesangial cells promotes interstitial fibrosis in DN. Expression changes in CASC2/miR135a-5p/TIMP3, CDKN2B/miR15b-5p/WNT2B, HOTAIR/miR147a/WNT2B, and NEAT1/miR-423-5p/GLIPR2 pathways lead to enhanced proliferation of mesangial cells and result in mesangial hypertrophy. Podocyte damage and renal function decrease were associated with MIAT/miR-130b/SOX4, Dlx6-AS1/miR346/GSK-3β, and Glis2/miR-328/Sirt1 [154]. Tubular epithelial cell injury and death were induced by ANRIL/miR-497/TXNIP, MALAT/miR-23c/ELAVL1, NEAT1/miR-34c/NRLP3, CDKN2B-AS1/miR98-5p/NOTCH2, and ZFAS1/miR-525/SGK1 signaling pathways [154].

According to a systematic review and meta-analysis of human case–control or cohort studies on differential lncRNA expression, 18 lncRNAs validated both in animals and humans in DN were as follows: CYP4B1-PS1–001, SOX2OT, HOXB3OS, NEAT1, GAS5, XIST, Lnc-ISG20, CASC2, PVT1, OIP5–AS1, H19, MEG3, NONHSAG053901, TUG1, CES1P1, MIAT, ANRIL, and MALAT1 [52]. Four lncRNAs from the list—NEAT1, CASC2, MALAT1, and ANRIL—have been validated in diabetic cells, tissues, and blood [52].

A recent review by Yu et al. emphasizes the modulation of lncRNA, regulating autophagy during DN progression [107]. lncRNAs AK044604, SPAG5-AS1, SNHG17, NEAT1, and Linc279227 were reported to be upregulated in DN mouse models and HG-treated podocytes, thus promoting podocyte apoptosis and kidney injury. In contrast, Gm5524/Gm15645, SOX2OT, Hoxb3os, and XIST enhance the viability of podocytes and reduce kidney fibrosis [107].

A number of studies indicate that circRNAs take part in DN development and can serve as biomarkers for monitoring DN [96,104,155]. Although most circRNAs have negative effects on DN, certain circRNAs have protective effects against DN. Decreased circRNA_010383 increases proteinuria and renal fibrosis, while elevated Circ_0125310 and Circ_DLGAP4 promote mesangial cell proliferation and fibrosis. CircHOMER1 increases oxidative stress, inflammation, and ECM deposition [104,155]. Evidence for the roles of circRNAs in DN mainly comes from mesangial cells (MCs), tubular epithelial cells (TECs), and podocytes [94]. For example, an investigation by Wang et al. (2021) in a mouse DN model and HG-exposed mesangial cells (MCs) has demonstrated that the circ_0037128/miR-17-3p/AKT3 axis significantly suppressed cell proliferation and fibrosis in DN pathogenesis [156]. The results published by Hu et al. suggested that the abnormal expression of the circRNA_15698/miR-185/TGF-β1 axis in HG-exposed MCs has a role in diabetic renal fibrosis [157]. Mou et al. confirmed 18 upregulated circRNAs and 22 downregulated circRNAs in the DN kidney from db/db mice using circRNA-seq [47]. Furthermore, circ_0000491 levels were significantly augmented in DN mice and HG-induced mouse MCs. In addition, circ_0000491 sponged miR-101b and activated TGFβR1, leading to ECM accumulation [47]. Upregulation of circ_0000491 in SV40-MES13 cells by HG was found to stimulate the expression of Hmgb1, which is a target of miR-455-3p. Knockdown of Circ_0000491 suppressed HG-induced apoptosis, inflammation, oxidative stress, and fibrosis by regulating the miR-455-3p/Hmgb1 axis [158]. Accumulating data demonstrate that circRNAs are associated with tubular epithelial cell damage in DN. Wen et al. explored the circRNA expression profiles and found that circular RNA actin-related protein 2 (circACTR2) was upregulated in HG-exposed HK-2 cells and mediated inflammation and pyroptosis [159]. Knockdown of circACTR2 prevented HG-induced pyroptosis, inflammation, and fibrosis of TECs, suggesting that circACTR2 has a vital role in the pathogenesis of DN [159]. Circular RNA of eukaryotic translation initiation factor 4 gamma 2 (circEIF4G2) was elevated in db/db mice and in the HG-induced NRK-52E rat kidney cells [160]. By contrast, the downregulation of circEIF4G2 mitigated renal fibrosis in DN by sponging miR-218 [160]. Recent evidence also suggested that circRNAs are involved in the injury and apoptosis of podocytes in DN. Yao et al. found that circ_0000285 was increased in kidney tissues of mouse models of DN and podocytes exposed to HG, leading to inflammation and podocyte injury through sponging miR-654-3p/mitogen-activated protein kinase 6 (MAPK6) [161].

## 5. Inflammation and Innate Immune Cells in T2DM: The Role of Non-Coding RNAs

T2DM is characterized by the development of chronic low-grade inflammation in the cardiovascular system, kidneys, adipose tissue, and other organs, in which monocyte/macrophages play the leading role [69,162,163]. In T2DM, monocytes/macrophages have increased ability to adhere to the vessel wall, to migrate through the endothelial layer, and to differentiate towards pro-inflammatory M1 macrophages, contributing to the development of macro- and microvascular complications [164,165,166]. Monocytes and macrophages control innate immune responses not only to pathogens, but also to the metabolic waste components using their broad arsenal of scavenger receptors [167,168,169,170]. They secrete inflammatory cytokines and reactive oxygen species, exacerbating endothelial dysfunction in vascular complications of diabetes, such as cardiomyopathy [171,172], nephropathy [165,173], and retinopathy [166,174]. Hyperglycemia is known to stimulate inflammatory reactions in monocytes/macrophages, and even to have at least middle-term pro-inflammatory memory linked to the increased expression of S100A9 and S100A10 that have damaging effects on endothelial cells [68,69,175,176,177]. Key signaling pathways regulating macrophage polarization include NF-κB, JAK/STAT, Notch, and TGF-β/SMAD [171]. However, there are still significant gaps in the understanding of the molecular basis of proinflammatory programming of monocytes and macrophages in T2DM, making further investigation urgently needed [172]. Single-cell sequencing (scRNA) of the total pool of peripheral blood mononuclear cells (PBMCs) of three patients with T2DM and three healthy donors provided new insight into immune status and potential immune mechanisms involved in the pathogenesis of T2DM. As a result, a transcriptional map of immune cells was created, 119 DEGs in T cells and 178 DEGs in monocytes circulating in the blood of patients with T2DM were identified [178].

### 5.1. miRNAs in Monocytes in T2DM

The study by Wang et al. explored the connections between T2DM and insulin resistance with expression of stress-related miR-18a and miR-34c using PBMCs of T2DM patients, subjects with impaired fasting glucose (IFG), and healthy individuals [179]. The increased levels of miR-18a and decreased levels of miR-34c in PBMCs were associated with the risk of T2DM and IFG. Expression levels of miR-18a and miR-34c were found to significantly correlate with chronic-stress-associated factors, such as cortisol, corticotropin-releasing factor (CRF), and IL-6 levels in the blood [179]. miR-18a and miR-34c in PBMCs may be important markers of stress reaction and may play a role in vulnerability to T2DM as well as IR [179].

Due to their phenotypic plasticity, macrophages play a role in both inflammation and resolution during cardiac and renal injury in T2DM. Studies demonstrated miRNAs expression changes in the context of monocyte differentiation and pro-inflammatory M1 macrophage activation in T2DM-associated chronic inflammation (Figure 2). miRNAs were found to be uniquely regulated in human macrophages polarized toward M1 and M2 [180]. miR-125a-3p and -193b, miR-27a-3p, miR-155-3p, and miR-29b-3p exhibited increased expression during macrophage activation, both in M1 and M2 phenotypes, while miR-26a showed decreased expression. miR-222-3p expression is uniquely upregulated in M2 macrophages but downregulated in M1 ones [181]. Distinct miRNA expression profiles were found in mice for M1 versus M2 macrophages, with miR-181a, miR-155-5p, miR-204-5p, and miR-451 being highly upregulated and miR-125-5p, miR-146a, miR-143-3p, and miR-145-5p significantly downregulated [182]. In mouse models of inflammation-associated T2DM complications, miR-155 was shown to be upregulated in M1 polarized macrophages [30]. The aggravation of diabetic cardiomyopathy caused by estrogen deficiency was prevented by treatment with antagomiR-155, which induced M2 macrophage infiltration and ameliorated the structure and function of the heart. It is suggested that miR-155 inhibition therapy could serve as a promising approach to improve cardiac function in DCM [183]. Under HG conditions, the expression of miR-126 in macrophages was downregulated, and overexpression of miR-126 diminished efferocytosis, phagocytic clearance of the apoptotic cells, and impairment in macrophages of DCM patients [30].

Increased expression of miR-99a in murine bone marrow-derived macrophages suppresses TNFα production, preventing M1 polarization, and reduces the level of proinflammatory markers in T2DM [184]. The therapeutic potential of miR-99a in db/db mice was evident due to improved glucose tolerance and insulin sensitivity after miR-99a mimics were administered [184]. Conversely, increased expression of miR-130b, miR-330-5p, and miR-495 enhances M1 polarization of peritoneal/adipose-tissue derived macrophages, which has been shown in murine models of high-fat diet induced T2DM [185,186,187]. Aggravated endothelial injury and impaired endothelial repair capacity were found to be associated with overexpression of miR-483-3p in the M2-type macrophages and in the endothelial cells of T2DM patients [188]. Overexpression of miR-483-3p increased endothelial and macrophage apoptosis and impaired reendothelialization in vitro [188]. Significant upregulation of miR-471-3p expression was detected in the diabetic mouse bone marrow-derived macrophages (db/db) and advanced glycation end products (AGE)-treated RAW264.7 cells [189]. Meanwhile, inhibition of miR-471-3p reduced proinflammatory macrophage polarization. Bioinformatic analysis identified *SIRT1* (coding NAD-dependent deacetylase sirtuin-1) as a target of miR-471-3p, which was then confirmed in a luciferase assay [189]. Recently, a therapeutic role was demonstrated for miR-369-3p in halting diabetes-associated atherosclerosis by regulating GPR91 and macrophage succinate metabolism [190]. In the study by Rawal et al., miR-369-3p was found to be reduced in peripheral blood mononuclear cells from diabetic patients with coronary artery disease (CAD) and in aortic intimal lesions from Ldlr-/- mice on a high-fat sucrose-containing diet. In vitro, oxLDL treatment reduced miR-369-3p expression in mouse bone marrow-derived macrophages (BMDMs) [190].

As mentioned above, miR-21 dysregulation was reported in T2DM-associated heart failure with preserved ejection fraction (HFpEF) [191]. A recent study investigated the effects of miR-21-3p on macrophage polarization and mitophagy in the chronic heart failure model of the isoproterenol (ISO)-induced myocardial structural disruption and fibrosis in rats. The ISO-challenged rats displayed increased left ventricular internal diameter systolic (LVIDs), decreased left ventricular ejection fraction (LVEF), which are traditional clinical indicators of HF, as well as elevated serum levels of creatine kinase-myocardial band, a marker of myocardial necrosis. In this model, heart failure was exacerbated by miR-21-3p [192]. miR-21-3p accelerated M1 macrophage polarization in vivo in a chronic heart failure model and in the co-culture system of macrophages and rat cardiomyocytes H9c2 cell line; furthermore miR-21-3p mimics promoted M1 polarization, which exacerbated H9c2 cell damage [192].

### 5.2. lncRNAs and circRNAs in Monocytes in T2DM

There is also multiple evidence for the role of lncRNAs and circRNAs in diabetes-related inflammatory processes [96]. For example, Reddy et al. showed that the long-ncRNA DRAIR plays an important role in reducing diabetes-related inflammation [193]. DRAIR suppression led to a decrease in the expression of anti-inflammatory genes in the THP1 monocyte-like cells, increased monocyte adhesion to the vessel wall, and the expression of pro-inflammatory genes such as *IL-1β* [193]. In contrast to DRAIR, lncRNA transcribed from the opposite strand of the dynamin 3 gene (*Dnm3os*) increased inflammatory responses that contribute to complications in T2DM. *Dnm3os* expression is significantly upregulated in CD14+ monocytes from different murine diabetes models, leading to the increased expression of proinflammatory genes in macrophages and increased phagocytic activity [194].

Transcriptome analysis of monocytes isolated from the peripheral blood of patients with T2DM and control healthy volunteers revealed two significantly elevated lncRNAs (ENSG00000287255 and ENSG00000289424) and one significantly decreased lncRNA (ENSG00000276603) in patients versus controls [49]. The most likely associated with T2DM were lncRNAs ENSG00000225822 (UBXN7-AS1), ENSG0000028654, ENSG00000261326 (LINC01355), ENSG00000260135 (MMP2-AS1), ENSG00000262097, and ENSG00000241560 (ZBTB20-AS1) [49]. Bioinformatic analysis of differentially expressed genes revealed the networks associated with cell mobility, growth, and development [49]. Yang et al. obtained RNA-seq data for PBMCs from three T2DM patients and three healthy individuals, which were used to construct a ceRNA network, demonstrating that lncRNAs and circRNAs play a regulatory role by interacting with multiple miRNAs, and, thus, co-regulating more mRNAs [20]. The authors also showed the important role of differentially expressed lncRNAs and circRNAs in T2DM patients in glucose metabolism, the mTOR signaling pathway, the lysosomal pathway, and the apoptotic pathway [20]. In the study by Ma et al. [42], 68 differentially expressed lncRNAs and 74 differentially expressed mRNAs were found using microarray analysis of the PBMC transcriptome of five T2DM patients and paired controls. Searching for lncRNA-mRNA interaction pairs revealed that the target gene of lncRNA XR_108954.2 encodes the transcription factor E2F2, which is involved in the control of the cell cycle, as well as glucose and lipid metabolism [42].

A comparative study of PBMC using RNA-seq revealed 220 circRNAs to be differentially expressed between patients with T2DM and healthy individuals [46]. circANKRD36 levels were upregulated in peripheral blood leukocytes and correlated with chronic inflammation in T2DM [46]. Another study analyzed the circRNA expression by RNA-seq in the polarized human THP-1 macrophage-like cells incubated in M1 (interferon-γ + LPS) or M2 (interleukin-4) conditions [48]. A total of 71 up-regulated circRNAs and 69 down-regulated circRNAs from 9720 detected circRNAs were found in the M1-type THP-1 cells. circRNF19B expression was significantly up-regulated in M1 conditions [48]. Bone marrow (BM) derived murine macrophages polarized to a pro-inflammatory phenotype (INFγ + TNFα) or anti-inflammatory phenotype (IL-10, IL-4, and TGF-β) were studied to determine whether circRNAs modulate monocyte/macrophage biology and function [195]. The authors used circRNA microarray analyses to assess transcriptome changes, which allowed them to choose circ-Cdr1as as playing a key role in regulating the anti-inflammatory phenotype of macrophages. Overexpression of circ-Cdr1as increased transcription of anti-inflammatory markers and percentage of CD206+ cells in naïve and pro-inflammatory macrophages in vitro [195].

To summarize, the accumulated published data indicate that dysregulated miRNA, lncRNA, and circRNA are extensively involved in the pathogenesis of T2DM and diabetes–induced cardio/renal complications and can act as key nodes in the disease regulatory network. NcRNAs associated with T2DM, demonstrating a potential as effective biomarkers, have been found in animal models, as well as in human studies. Unique as well as common biomarkers for patients and challenged rodents were found. Notably, despite intensive research in the field, there are still gaps in the understanding of the complex integrating molecular mechanisms of dozens of ncRNAs involved in the T2DM initiation and its major complications. Therefore, further investigations are urgently needed based on high-throughput RNA sequencing techniques, refined computational analysis, accumulated data integration, and construction of T2DM molecular networks, including external inducing signals, internal multiple signaling pathways, and target processes under the ncRNA regulation. Developing the strategy for integrative data analysis is necessary to evaluate potential ncRNA-based biomarkers for clinical practice, to provide novel targets for disease treatment, and new drug development.

## 6. Involvement of ncRNAs in the SGLT2-i Effect on Cardiac and Renal Complications of T2DM

### 6.1. SGLT2-i Effect on the Cardiovascular System State and Function

Canagliflozin, dapagliflozin, empagliflozin, ertugliflozin, and sotagliflozin are the five therapeutic agents of a class of SGLT2 inhibitors that have been approved to date by the FDA; they are now being considered as a second-line therapy for the management of T2DM [3,9,196,197]. SGLT2-is improve cardiovascular complications in T2DM through a combination of hemodynamic, metabolic, anti-inflammatory, and vascular effects on the myocardium [50]. These drugs reduce inflammation, oxidative stress, apoptosis, and lipotoxicity, enhance endothelial function, and promote a shift towards more efficient energy utilization. Studies have explored various cardiac cell types, including cardiomyocytes, endothelial cells, fibroblasts, and smooth muscle cells, as potential targets for the observed cardiovascular benefits of SGLT2 inhibitors [197]. Empagliflozin reduced cardiac fibrosis through modulation of human myofibroblast function and suppression of the expression of several pro-fibrotic factors, including collagen type I α1, actin alpha 2 (smooth muscle), connective tissue growth factor, fibronectin 1, and matrix metalloproteinase 2 [9,198]. Dapagliflozin inhibited JNK, p38, and FoxO1 signaling pathways in association with the reduction in myocardial interstitial and perivascular fibrosis. The downregulation of mitogen-activated protein kinase/activator protein-1 signaling by dapagliflozin has been found to be Na^+^/H^+^ exchanger 1 (NHE1)-dependent [199]. Dapagliflozin protected against myocardial fibrosis by inhibiting resident fibroblast activation, collagen secretion through suppression of EMT, and AMP-activated protein kinase α-mediated inhibition of TGF-β/Smad signaling [200]. Numerous studies have evidenced that empagliflosin directly inhibits Na^+^/H^+^ exchanger 1 (NHE1) activity in cardiac cells, including cardiomyocytes, endothelial cells, and cardiofibroblasts [201]. A recent study emphasized that dapagliflozin and/or liraglutide attenuated cardiac tissue injury via reducing key elements of oxidative stress, inflammation, and apoptosis in diabetes-induced cardiomyopathy rats [202]. Empagliflozin partially exerted anti-oxidative stress and anti-apoptotic effects on cardiomyocytes under high glucose conditions by activating AMPK/PGC-1α and suppressing other RhoA/Rho-associated protein kinase (ROCK) pathways in diabetes-induced cardiomyopathy mice [203]. In addition to inflammation and oxidative stress, lipotoxicity also plays a critical role in diabetic HFpEF. Sun et al. showed that canagliflozin can attenuate lipotoxicity and inflammatory injury in cardiomyocytes and can protect diabetic mouse hearts via inhibiting the mTOR/hypoxia-inducible factor-1α (HIF-1α) pathway [204].

SGLT2-is improved cardiac energetics and enhanced mitochondrial function by inhibiting cardiac glucose uptake and glycolysis, increasing cardiac glucose oxidation, ketone metabolism, and fatty acids metabolism [196,205]. Canagliflozin ameliorated myocardial remodeling in HFpEF rats by optimizing cardiac energy metabolism, enhancing mitochondrial function, and consequently reducing myocardial hypertrophy and fibrosis, with simultaneous improvement in diastolic function [206].

### 6.2. Effect of microRNA on the Cardiovascular System Condition and Functions Under the Treatment with SGLT2-i

Protective effects of SGLT2 inhibitors’ action in various states and conditions, including cancer, metabolic diseases, and others, by affecting ncRNAs expression were demonstrated in cultured cells, animal models, and patient studies for a number of SGLT2 inhibitors [8]. The involvement of non-coding RNAs in the SGLT2 inhibitors-induced cardioprotective effects in T2DM was demonstrated for empagliflozin and dapagliflozin (Figure 2; Table 2).

#### 6.2.1. miR-21 and miR-92 in the Effect of SGLT2-i

miR-21 plays a pivotal role in cardiac function regulation; therefore, it has been frequently investigated, taking into account its prospects in clinical therapy. This miRNA is significantly upregulated in conditions such as coronary arterial disease (CAD), cardiac fibrosis (CF), and heart failure (HF) in T2DM patients [207,208,209]. In contrast, according to Tao L, the circulating miR-21 levels were significantly decreased in diabetic cardiomyopathy patients compared to the non-DCM group [210]. The disparate findings on miR-21 in cardiomyopathy suggest that its pathological activity depends on the cell type and on the disease context. Based on the results of the bioinformatics study by Sessa et al., the role of miR-21 in cardiovascular diseases is related to its interaction with a number of hub genes: programmed cell death 4 (*PDCD4*), RAS guanyl-releasing protein 1 (*RASGRP1*), and B-cell translocation gene 2 (*BTG2*) (bioinformatics analysis of the experimental data from CVD patients) [211]. miR-21 is associated with elevated reactive oxygen species (ROS) levels and is involved in endothelial dysfunction. It targets key genes involved in the regulation of ROS production, such as Krev/Rap1 interaction trapped-1 *(KRIT1*) and superoxide dismutase 2 (*SOD2*), resulting in increased oxidative stress when overexpressed [212]. Experiments in the cellular and animal models, as well as in patients, indicate that miR-21 facilitates the accumulation of superoxide in cells, particularly in the context of high glucose concentrations, which is particularly relevant in diabetic vascular complications [212].

Ridwan et al. [213] in the Wistar rat model of STZ-induced hyperglycemia found a trend of decreased miR-21 expression in the heart under the influence of empagliflozin compared to metformin (Table 2). The authors associate this effect of SGLT2-i with the detected decrease in the TGF-β1 level and the increased expression of the matrix metalloproteinase-2 (*MMP-2*) gene, which may reduce the risk of cardiac fibrosis occurrence during treatment with empagliflozin, but not metformin [213]. It should be noted that in an earlier study in a STZ/HFD-induced mouse model of diabetic cardiomyopathy, increased miR-21 expression was associated with promoted fibrosis, decreased autophagy, and attenuated the protective effects of the hypoglycemic drug vildagliptin, a dipeptidyl peptidase 4 inhibitor, on cardiac function [214]. In this study, miR-21 was also upregulated in the rat cardiomyocyte cell line H9c2 under high glucose conditions, which was downregulated by vildagliptin. The authors showed that pathological changes in the heart under hyperglycemia occur as a result of autophagy reduction through the SPRY1/ERK/mTOR signaling pathway, the activity of which is suppressed by increased levels of miR-21, and normalized by the vildagliptin therapy [214].

Mone et al. assessed the effect of empagliflozin on the expression of circulating microRNAs involved in the regulation of endothelial function in a group of age-related T2DM patients with heart failure with preserved ejection fraction (HFpEF) [215]. The authors analyzed by RT-PCR a panel of microRNAs previously reported to be involved in the regulation of endothelial dysfunction [215]. Compared with the control group, significantly increased levels of circulating miR-21 and miR-92 were found in HFpEF patients [215]. After 3 months of empagliflozin treatment, circulating miR-21 and miR-92 levels in T2DM patients with HFpEF decreased, while they did not change in response to metformin or insulin [215]. As noted above, both microRNAs have been previously associated with inflammatory diseases, in particular, heart failure, and may have value as prognostic markers for cardiomyopathy in T2DM. miR-92a has been found to inhibit heme oxygenase-1 (HO-1), an important enzyme responsible for the degradation of heme to form biliverdin and bilirubin, which have antioxidant properties [216]. When miR-92a was inhibited, HO-1 levels increased, thereby enhancing the antioxidant potential of human umbilical vein endothelial cells and db/db mouse aortas [216].

Notably, in the obese T2DM patients, abdominal fat excision reduced inflammation, normalized left ventricular diastolic function over 1 year of follow-up; these improvements were accompanied by decreased miR-21 and miR-92 and increased miR-126 levels in the blood serum [217]. Previously, miR-126 was shown to be involved in the protection of the cardiovascular system by modulating angiogenesis, inflammation, and apoptosis in patients with T2DM, with a number of studies establishing a decrease in the level of this microRNA in the blood serum of patients with cardiomyopathy and T2DM [53,139]. It was shown that miR-126 is expressed at significantly lower levels in the heart of diabetic rats, which is associated with diastolic heart failure [218]. This microRNA is a potential target for investigating the anti-inflammatory effect of SGLT2 inhibitor drugs.

As mentioned above, increased miR-21 and decreased miR-126 were found to induce the pro-inflammatory M1 macrophage phenotype in cardiac T2DM disorders; therefore, one can propose a macrophage pivotal role in the SGLT2-i-induced patients’ health improvements [37,192]. This question has not been examined yet and needs further macrophage-targeted investigation.

#### 6.2.2. miR-30d in the Effect of SGLT2-i

In the work by Zhang et al. the therapeutic effect of SGLT2 inhibitors (not specified) on the development of diabetic cardiomyopathy was studied in a rat model on a high-fat diet and with streptozotocin (STZ)-induced diabetes [219]. The use of SGLT2 inhibitors for 10 weeks improved cardiac function as a result of enhancing autophagy by reducing the expression level of miR-30d, which was significantly increased during the development of DCM in this model [219]. In addition, in the case of the miR-30d knockdown, cardiac function was also improved. The study showed that the SGLT2 inhibitor improves the animal condition with diabetic cardiomyopathy by affecting the miR-30d/KLF9/VEGFA pathway, which regulates the expression of autophagy factors protein light chain 3 (LC3-II) and p62/SQSTM1 (p62/sequestosome 1) [219]. Earlier, Li et al. (2014) also found an increase in miR-30d levels, which contributed to cardiomyocyte pyroptosis in rats with streptozotocin (STZ)-induced diabetes, as well as in cardiomyocytes treated with a high level of glucose [220]. Pyroptosis is one of the types of programmed cell death, which is accompanied by an inflammatory response, playing a significant role in the pathogenesis of cardiomyopathy. Li et al. showed that the induction of cardiomyocyte pyroptosis under hyperglycemia under the influence of increased miR-30d expression occurs through an increase in the level of caspase-1 and proinflammatory cytokines IL-1β and IL-18, as a result of a decrease in the expression level of foxo3a and its direct target activity-regulated cytoskeleton-associated protein (ARC) [220].

El Khayari et al. reviewed the important role of miR-30d and other members of the miR-30 family (particularly miR-30a and miR-30e) in the development of T2DM complications in the heart and kidneys [196]. A recent review has summarized the data obtained in the cellular and animal models, as well as in patients, and described the mechanisms underlying the effect of SGLT2 inhibitors and miR-30 family members on endothelial cells and cardiomyocytes. The mechanistic elements targeted by SGLT2 inhibitors and miR-30 family members included transcription factor 21 (TCF21), Toll-like receptor 4 (TLR-4), AMP-activated protein kinase (AMPK), and angiopoietin 2, followed by suppression of nuclear factor-kB (NF-κB) signaling and decreased expression of proinflammatory cytokines and adhesion molecules [196]. Given the mutual influence of SGLT2 inhibitors and miR-30 on inflammation, angiogenesis, and endothelial cell survival, the critical role of the interaction between different cell types, including immune cells, seems highly likely. Thus, members of the miR-30 family may be involved in mediating the action of SGLT2 inhibitors in endothelial cells and cardiomyocytes [196]. However, this was not directly demonstrated, and such interaction in immune cells cannot be excluded.

#### 6.2.3. miR-181a and Empagliflozin

Empagliflozin had the potential to ameliorate MI-induced myocardial fibrosis by inhibiting the expression of miR-181a, suppressing the TGF-β1/Smad3 signaling pathways in the myocardial tissue [221]. miR-181a was earlier associated with CAD development in T2DM, both in patient studies and animal CAD models, and HG-induced endothelial cells [187]. Moreover, a study by Chen et al. demonstrated that miR-181a upregulation leads to the downregulation of HMGB2 in the angiotensin II-treated primary cardiomyocytes (ex vivo model of cardiac hypertrophy) [222].

#### 6.2.4. miR-193b and Empagliflozin

Molecular mechanisms of empagliflozin effects were studied in the obese ZSF-1 leptin-receptor knockout rats (heart failure with preserved ejection fraction mode, HFpEF l) and in the obese ZSF-1 rats treated with SU5416 to stimulate resting pulmonary hypertension (obese + sugen, CpcPH model) [223]. In heart failure with preserved ejection fraction (HFpEF), the metabolic syndrome promoted pulmonary vascular dysfunction and exercise-induced pulmonary hypertension (EIPH) by enhancing ROS production and increasing miR-193b levels. This factor had a suppressive effect on the nuclear factor Y α subunit (NFYA)-dependent soluble guanylate cyclase β1 subunit (sGCβ1) expression in the pulmonary arterial smooth muscle cells [223]. Application of empagliflozin in the CpcPH rat model with combined precapillary and postcapillary pulmonary hypertension ameliorated metabolic syndrome and reduced mitochondrial reactive oxygen species generation downregulates miR-193b, resulting in restoration of NFYA-sGCβ1-cGMP signaling and improvement in EIPH symptoms [223]. miR-193b was recently identified as a potential marker of T2DM in a large-scale analysis of 1934 miRNAs circulating in the blood plasma of newly diagnosed, drug-naive patients [224]. When miR-193b-3p levels were increased, triosephosphate isomerase (TPI1) levels were decreased in both the plasma of T2DM patients and HepG2 hepatocellular carcinoma cells [224]. The multi-omics approach revealed that miR-193b-3p may affect glucose metabolism by directly targeting the tyrosine 3-monooxygenase/tryptophan 5-monooxygenase activation protein zeta (YWHAZ)/14-3-3ζ and upregulating the transcription factor forkhead box protein O1 (FOXO1), which is involved in the effects of the PI3K-AKT signaling pathway. Based on the results observed in cultured HepG2 cells derived from liver tissue, it can be speculated that the effects of miR-193b-3p on glucose metabolism in other tissues may be similar [224]. The involvement of differential expression of miR-193b-3p in the development of cardiovascular complications of T2DM and in the implementation of the protective effect of SGLT2 inhibitors seems promising for further investigation.

#### 6.2.5. miR-30e-5p and miR199a-3p in the Effect of Dapagliflozin

In the study of Solini et al., patients with T2DM and hypertension (blood pressure >130/80 mm Hg) were randomly divided into two groups receiving either dapagliflozin or hydrochlorothiazide (HCT) during 4 weeks; their background medication was unchanged during the study [225]. The authors assessed the effect of short-term treatment on endothelial function and systemic and renal vascular function using standard, generally accepted instrumental approaches in association with changes in the expression of circulating microRNAs. Dapagliflozin, but not HCT, significantly altered the expression of two microRNAs previously associated with heart failure (miR-30e-5p and miR-199a-3p), suggesting the involvement of these microRNAs in the therapeutic effect of dapagliflozin [225]. As demonstrated earlier, miR-30e-5p, which is selectively up-regulated by dapagliflozin, inhibits myocardiocyte autophagy through modulation of the angiotensin-converting enzyme 2 (ACE2) pathway in doxorubicin-induced heart cardiomyopathy [226]. Increased expression of miR-30e-5p was associated with antihypertrophic and antiapoptotic effects on angiotensin II (Ang-II)-stimulated hypertrophic cardiomyocytes [227]. In contrast, miRNA-199a is a prohypertrophic miRNA, and its overexpression in mouse cardiomyocytes induced hypertrophy and suppressed autophagy [228]. The reduced expression of miR-199a-3p induced by dapagliflozin might also be intriguing in mechanistic terms, as far as antagonizing miR-199-3p could derepress cardiac peroxisome proliferator-activated receptor delta (PPARδ) levels, thus restoring mitochondrial fatty acid oxidation and improving cardiac function in the failing heart [229].

#### 6.2.6. Potential Role of miR-141 in the Effect of SGLT2-i

Li et al. demonstrated that SGLT2 mRNA is a direct target of miR-141, so miR-141 overexpression reduced the development of cardiac fibrosis in a rat myocardial infarction model by reducing the expression level of SGLT2 [230]. The study revealed an increase in the expression level of SGLT2 in association with a decrease in the expression of miR-141 during the development of fibrosis after the induction of myocardial infarction in rats via the permanent ligation of the left anterior descending branch of the coronary artery, and SGLT2 knockdown improved the condition of the heart muscle, as did an increase in miR-141 expression [230]. The results of this work open the question about possible synergistic effects of SGLT2 inhibitors and miR-141 in the anti-fibrotic effect in cardiomyopathy.

### 6.3. SGLT2-is Effect on the Renal System

Protective effects on renal function, which is impaired during the development of T2DM, have been demonstrated in cultured cells and in vivo models for several SGLT2 inhibitors [3,8,231,232]. A literature review by Castoldi et al. summarized the data demonstrating that SGLT2 inhibitors also effectively improve kidney state and function in various non-diabetic kidney disease models [233].

In the diabetic kidney disease models, SGLT2 inhibitors like dapagliflozin, empagliflozin, and canagliflozin have shown significant anti-inflammatory, anti-oxidative, and anti-fibrotic properties. They reduced calcification in vascular smooth muscle cells, inhibited epithelial–mesenchymal transition, and delayed fibrosis by modulating pathways such as YAP/TAZ, TXNDC5, and NLRP3 [8,231,234]. Dapagliflozin, empagliflozin, and canagliflozin have demonstrated pleiotropic vascular protective effects beyond the glycemic control, with benefits observed both in patients and animal models. These agents mitigate mitochondrial ROS production, suppress NOX4 expression, activate the AMPK/SIRT1/PGC-1α axis, preserve mitochondrial membrane potential, and enhance eNOS phosphorylation. All these mechanisms contribute to the improvement in the endothelial function and reduction in arterial stiffness [232,235]. Furthermore, SGLT2 inhibitors impact kidney function by regulating cell migration, reducing oxidative stress, and mitigating the effects of high glucose levels on the proximal tubule cells, demonstrating high potential as therapeutic agents for diabetic nephropathy [231]. SGLT2 inhibitors have been shown to impact kidney proximal tubule cell migration by modulating various cellular pathways involved in cell motility [236]. Empagliflozin exerted the nephroprotective effects in high-glucose-induced cultured kidney proximal tubule cells in vitro [237]. Dapagliflozin may delay tubulointerstitial fibrosis in diabetic kidney disease, inhibiting YAP/TAZ activation in tubular epithelial cells [238]. Anti-fibrotic and anti-ferroptotic effects of dapagliflozin in the DN mouse model have been associated with its interaction with the Nrf2 and TGF-β signaling pathways [234]. A recent whole-transcriptome study of the kidney tissue by Wu et al. indicated that empagliflozin can improve the outcome of DN in the mouse T2DM model by inhibiting inflammation, apoptosis, senescence, and other related pathways [239]. The KEGG enrichment of upregulated genes in the DN and DN treated with empagliflozin groups was concentrated in the p53 pathway, which was related to apoptosis and senescence, and the aggregated peroxidase proliferators activate receptor-related (PPAR) pathway, which was the main inflammatory pathway [239].

### 6.4. Effect of microRNA and lncRNA on the Renal System Condition and Functions Under SGLT2-is Treatment

Protective effects on the impaired renal function in T2DM were demonstrated in cultured cells, animal models, and patient studies for SGLT2 inhibitors canagliflozin, dapagliflozin, empagliflozin, irtugliflozin, and sotagliflozin. The involvement of several non-coding RNAs in the SGLT2 inhibitors-induced nephroprotective effects was demonstrated for empagliflozin and dapagliflozin (Figure 2; Table 2).

#### 6.4.1. miR-21 in the Effect of DN Therapy by Empagliflozin and Dapagliflozin in Different Models

Empagliflozin exerted a protective effect on human renal proximal tubular cells (HK-2) cultured in the presence of high glucose in vitro [237]. Empagliflozin attenuated HG-induced oxidative stress, matrix metalloproteinase (MMP2) activation, and RECK suppression, which led to stimulation of fibrosis in the kidneys of db/db mice by activating EMT and renal proximal tubular cells migration [237,240]. Das et al. showed that in HG-treated HK-2 cells, the advanced glycation end products (AGE) activated their receptor RAGE, causing the increased expression of TRAF3-interacting protein 2 (TRAF3IP2). Next, through the activation of NF-κB, p38MAPK led to an increase in the expression of miR-21 and suppression of the RECK expression, which is a direct target of miR-21 [237]. Thus, downregulation of miR-21 expression is one of the pathways mediating the therapeutic effect of empagliflozin on the renal proximal tubular cells in vitro. Recently, Al-Tantawy et al. studied the antidiabetic effect of the treatment by dapagliflosin in combination with gold nanoparticles (AuNPs) in the streptozotocin-induced rat model of diabetic nephropathy [241]. The treatment for 7 weeks significantly increased catalase activity and serum total antioxidant capacity, along with a substantial decline in malondialdehyde, indicating the alleviation of DN [241]. Autophagy activation, inhibition of apoptosis, and renal fibrosis were detected in renal tissue, which could be due to their inhibitory impact on miR-192 and miR-21 renal expression [241].

The protective effect of dapagliflozin was examined in the gentamicin-induced renal injury in Wistar rats, and improvement in kidney function and structure was found to be associated with an increase in miR-21 and a decrease in miR-181a levels in the kidney tissue [242]. Dapagliflozin administration decreased markers of kidney injury, such as increased creatinine, blood urea, and urine protein. The authors suggest that miR-21 and miR-181a expression changes in the kidney tissue by dapagliflozin lead to a decrease in renal tubular cell apoptosis [242]. Thus, SGLT2 inhibitors normalize abnormal miRNA expression in renal tissue in different pathological states: both chronic metabolic stress, associated with increased miR-21 levels, and acute drug-induced injury associated with decreased miR-21 levels.

#### 6.4.2. Exosomal miR 27a-3p, 190a-5p and 196b-5p as Therapy Efficacy Markers

Since ncRNAs appear to be promising predictive biomarkers for therapy efficacy, there is great interest in identifying novel differentially expressed ncRNAs using large-scale analysis. To assess the pharmacological effects of empagliflozin therapy on the kidneys, Shepard used a mouse model of early T2DM associated with obesity and insulin resistance [243]. The TallyHo/Jng mouse line, derived from Swiss-Webster, is a polygenic model of progressive insulin resistance leading to the development of T2DM on a high-fat diet [243]. The kidney function was assessed using a number of criteria, including urine microRNA expression profiling using large-scale microarray analysis, and changes in several identified microRNAs were verified by PCR analysis [243]. Empagliflozin treatment resulted in improved renal function parameters and changes in urinary exosomal microRNA profiles, including increased levels of miR-27a-3p, miR-190a-5p, and miR-196b-5p. Network analysis identified “cancer pathways” and “FOXO signaling” as the main pathways regulated by the drug [243]. An earlier study by An et al. found that increased levels of another member of the miR-196 family (miR-196a) in the urine of patients with diabetic nephropathy were a prognostic marker for kidney disease progression [244]. Four-week treatment with dapagliflozin of patients with hypertension and T2DM demonstrated a beneficial effect on the ability of the kidneys to dilate blood vessels in association with significant upregulation of circulating miR-30e-5p and downregulation of miR-199a-3p, which are involved in the pathophysiology of CV disease and heart failure [225]. Of note, the nephroprotective effect correlated with the pretreatment expression levels of circulating miR-27b and miR-200b [225].

#### 6.4.3. lncRNA, circRNA, and miRNA Identification and Study in DN Treatment with Dapagliflozin and Empagliflozin

The effect of dapagliflozin was studied using genome-wide sequencing and ceRNA network construction in a diabetic nephropathy (DN) model (db/db mice), which allowed comparative analysis of the expression of lncRNAs and mRNAs in the kidney tissue of healthy mice, db/db mice, and dapagliflozin-treated db/db mice [245]. db/db mice were characterized with glucose intolerance, high urinary albumin/creatinine ratio, and renal damage, which were found to be improved in the dapagliflozin group after 12-week treatment. Large-scale RNA sequencing identified 172 differentially expressed lncRNAs between DN and control mice, and 40 differentially expressed lncRNAs between DN without and with dapagliflozin. In these two pairwise comparisons, several common differentially expressed lncRNAs were found (NR_015554.2, XR_382492.3, XR_382493.3, XR_382494.3, XR_388840.1, XR_873495.2, and XR_876705.2).

The results of quantitative RT-PCR showed that dapagliflozin could reverse the expression changes in XR_382492.4, XR_873495.3, XR-388840.1, NR-015554.2, XR-382493.3, and XR-876705.2 associated with diabetic nephropathy [245]. The authors used the results of RNA sequencing and network pharmacology analysis of therapeutic pathways and targets to identify potential mechanisms of the dapagliflozin effect in DN. As a result, the involvement of TGF-β signaling pathways, peroxisome proliferator-activated receptor (PPAR) signaling pathway, chemokine signaling pathway, and a number of other pathways that affect 11 key genes (*SMAD9*, *PPARG*, *CD36*, *CYP4A12A*, *CYP4A12B*, *CASP3*, *H2-DMB2*, *MAPK1*, *MAPK3*, *C3*, *IL-10*) in the action of dapagliflozin was revealed [245]. It should be noted that the comprehensive analysis of the results revealed for the first time the involvement of the *SMAD9*, *CYP4A12a*, *CYP4A12b*, and *H2-DMB2* genes in DN, while these genes were previously unknown potential targets for dapagliflozin in DN [245].

Recently, Wu et al. conducted a comprehensive study of the mouse transcriptome in a model of induced diabetic nephropathy in mice treated with streptozocin to determine the dynamics of various types of ncRNAs and construct a ceRNA network of the response to empagliflozin [239]. They experimentally identified differentially expressed mRNAs in kidney tissue as a result of two pairwise comparisons: between the negative control and diabetic nephropathy, and between DN and DN with empagliflozin administration. A total of 232 genes were found downregulated and 282 genes upregulated for both comparisons. The authors focused on genes encoding proteins involved in metabolism, for which protein–protein interaction networks were constructed, while all data on lncRNAs and miRNAs that could potentially interact with them were obtained from the corresponding databases [239]. Construction of the ceRNA network allowed the identification of the β-1,4-mannosyl glycoprotein 4-β-n acetylglucosamine transferase (*Mgat3*), ST8α-n-acetylceramide α-2,8-sialic transferase 1 (*St8sia1*), and glucosamine-6-phosphate deaminase 1 (*Gnpda1*) genes as central ones. The role of the associated key microRNAs miR-466i-3p, miR466f-3p, miR-709, miR-207, and miR-297b-3p in DN and in the empagliflozin effects was not recognized and requires further investigation [239].

Song et al. studied the mechanisms of dapagliflozin action on cultured human renal proximal tubular epithelial cells HK-2 under HG conditions [246]. The results obtained using RNA microarray hybridization were used to construct ceRNA networks and functional analysis, which revealed the involvement of multiple circRNAs, their target miRNAs, and proteins in the action of dapagliflozin on renal proximal tubular cells. The experimental validation identified the hsa_circRNA_012448-hsa-miR-29b-2-5p-GSK3β pathway involved in oxidative stress response as an important pathway mediating the action of dapagliflozin in the treatment of diabetic kidney disease [246]. This finding is in accordance with the data from Wang et al., which demonstrated that miR-29 exhibited an anti-fibrotic effect in the renal tissue [247].

### 6.5. Effect of SGLT2 Inhibitors on Immune Cells in T2DM: Involvement of ncRNAs

One of the key mechanisms of SGLT2 inhibitors that underlay their cardioprotective and renoprotective effects in patients with T2DM is a reduction in chronic low-grade inflammation [231,248]. Recently, Shen et al. performed high-throughput analysis and demonstrated that pathways related to immune response and metabolism pathways were critically involved in diabetic nephropathy progression and reversed by dapagliflozin [249] (Table 2). ATAC-seq, RNA-seq, and weighted gene correlation network analysis have been performed to assess the chromatin accessibility in combination with mRNA expression in the comparative study of kidney tissue from the mouse diabetic nephropathy model: db/m mice (Controls), db/db mice (DN), and db/db mice treated with dapagliflozin (DN treatment group) [249]. Combined analysis of the data identified some candidate genes (including UDP glucuronosyltransferase 1 family, *Dock2*, *Tbc1d10c*) and transcriptional regulators (*KLF6* and *GFI1*) that might be associated with DN and dapagliflozin restoration [249]. However, the specificity of the action of these genes in the distinct cell types was not addressed. In particular, it would be of interest to examine the mechanistic role of *KLF6* in kidney macrophages. There are data indicating that *KLF6* is involved in the macrophage polarization and non-coding RNA regulates the KLF6 pathways in macrophages. For example, miR-21-5p carried by adMSC-derived apoptotic bodies was shown to bind to *KLF6*, leading to M2 polarization in skin wound healing [250]. In liver fibrosis induced by carbon tetrachloride in mice, miR-148a regulated intrahepatic macrophage functions through KLF6/STAT3 signaling [251]. Another study used the mode for the sterile inflammation kidney based on the ischemia/reperfusion-induced acute kidney injury (IRI-AKI). In this model, formononetin (a natural, O-methylated isoflavone of plant origin) significantly improved renal function and reduced inflammation in IRI-AKI, and the authors attributed these effects to the inhibition of KLF6/STAT3-mediated macrophage pro-inflammatory activation [252].

Intensive investigation of molecular mechanisms of SGLT2 inhibitors action on macrophages, key innate immune cells, suggests that the anti-inflammatory effect of inhibitors relies on the attenuation of inflammasome activity and inhibition of TLR4/NF-κB [231,248]. The anti-inflammatory effect is also related to the changes in the activity of other signaling pathways (AMPK, PI3K/Akt, ERK 1/2-MAPK, and JAKs/STAT) in macrophages [231]. A recent study investigated the inhibitory effect of dapagliflozin on macrophage M1 polarization and apoptosis in the murine model of streptozotocin-induced diabetic nephropathy [253]. Experiments ex vivo demonstrated that dapagliflozin attenuated M1 macrophage polarization and mitigated apoptosis in the abdominal macrophages of diabetic mice by suppressing the phosphoinositide 3-kinase (PI3K)/protein kinase B (AKT) signaling pathway (PI3K, AKT, phosphorylated protein kinase B) [253]. However, the role of ncRNAs in SGLT inhibitors’ effects on the innate immune cells remains poorly investigated, and only isolated reports are available.

Abdollahi et al. studied the effect of dapagliflozin on human macrophages differentiated out of peripheral blood mononuclear cells and activated by LPS [254]. Addition of dapagliflozin reduced the M1/M2 ratio and LPS-induced TLR-4 overexpression in macrophages under high and low glucose conditions and also suppressed the high glucose-induced activation of the NF-κB signaling pathway. Dapagliflozin treatment abolished the LPS-induced change in the expression of two microRNAs, miR-146a (anti-inflammatory) and miR-155 (pro-inflammatory), both are known to play a role in the regulation of inflammation through activation of the NF-κB signaling pathway [254]. Earlier, in a model of STZ-induced DN in wild-type mice, miR-146a expression increased in both peritoneal and intrarenal macrophages, playing an anti-inflammatory role [255]. Also, miR-146a deficiency during STZ-induced diabetic nephropathy in miR-146a−/− mice led to the increased expression of M1 activation markers and to the suppression of M2 markers in renal macrophages. Mechanistically, miR-146a suppresses expression of its target genes *Irak1* and *Traf6* in renal macrophages. IRAK1 and TRAF6 are known to promote the NFκB-mediated upregulation of proinflammatory cytokines, such as IL-1β, IL-18, and TNF-α, inhibiting the inflammatory response [255]. miRNA-155 is also involved in innate immune response, playing a pro-inflammatory role [79]. The overexpression of miR-155 promoted the activation of the ox-LDL-induced NLRP3 inflammasome in PMA-preprocessed THP-1 macrophages, thus aggravating the carotid atherosclerotic lesion in ApoE−/− mice [256]. Repressing the expression of miR-155 while promoting that of miR-146a, dapagliflozin can effectively modulate inflammatory responses preceding T2DM-induced vascular complications development.

**Table 2 ijms-26-11198-t002:** NcRNAs as modulators of therapeutic effects of SGLT2 inhibitors: focus on kidney and heart.

Specific Inhibitor/Kidney or Heart	ncRNA	Mouse/Rat Model, Patients’ Cohort	Effect of Specific Inhibitor	Major Effect in Cells or in Circulation	Ref
Empagliflozin/Heart	miR-21	Rat model of STZ-induced hyperglycemia	Empagliflozin treatment leads to decreased TGF-β1 level/increased expression of the matrix metalloproteinase-2 (MMP-2) in heart tissue	Empagliflozin administration for seven weeks decreases (compared to metformin) miR-21 levels in heart tissue	[213]
Empagliflozin/Heart/ HFpEF	miR-21 miR-92	Age-related T2DM patients with heart failure with preserved ejection fraction (HFPEF)	Empagliflozin improves endothelial function by reducing mitochondrial calcium overload and the generation of reactive oxygen species	Empagliflozin (not metformin) decreases biomarkers of HF: circulating miR-21 and miR-92 levels in T2DM patients with HFPEF	[191,215]
SGLT2 inhibitor (not specified)/Heart/DCM	miR-30d	Rat model of diabetic cardiomyopathy (DCM) on a high-fat diet and with (STZ)-induced diabetes	SGLT2 inhibitor (not specified) improved cardiac function due to enhancing autophagy by reducing the expression level of miR-30d	SGLT2-i improves DCM by affecting the miR-30d/KLF9/VEGFA pathway, which regulates the expression of autophagy factors protein light chain 3 (LC3-II) and p62/SQSTM1 (p62/sequestosome 1)	[219]
Empagliflozin/Heart/HFpEF	miR-193b	Rat model of combined precapillary and postcapillary pulmonary hypertension (CPCPH): obese ZSF-1 rats treated with SU5416 to stimulate resting pulmonary hypertension	Empagliflozin ameliorates metabolic syndrome and reduces mitochondrial reactive oxygen species generation	Empagliflozin downregulates miR-193b, resulting in restoration of NFYA-SGCβ1-cgmp signaling in pulmonary arterial smooth muscle cells and improvement in EIPH symptoms	[223]
Dapagliflozin/Heart/TD2M, hypertension	miR-30e-5p miR-199a-3p	Patients with T2DM and hypertension (blood pressure >130/80 mm Hg)	Short-term treatment (4 weeks) by dapagliflozin improves endothelial function, systemic vascular function	Dapagliflozin increases circulating levels of antihypertrophic miR-30e-5p, decrease in circulating levels of prohypertrophic miR-199a-3p, which are involved in the pathophysiology of CV disease and heart failure	[225]
Dapagliflozin/Kidney/Renal vascular function	miR-27b miR-200b	Patients with T2DM and hypertension (blood pressure > 130/80 mm Hg)	Dapagliflozin demonstrates beneficial effect on the kidneys’ ability to dilate blood vessels	Dapagliflozin’s nephroprotective effect correlates with the pretreatment levels of circulating miR-27b and miR-200b	[225]
Empagliflozin/Kidney	miR-21	High glucose (HG) treated human kidney proximal tubule cells (HK-2)	Empagliflozin treatment ameliorated HG-induced inflammation and inhibited epithelial-to-mesenchymal transition (EMT) and HK-2 cell migration	Empagliflozin ameliorates AGE–induced RECK expression suppression via oxidative stress/TRAF3IP2/NF-κb and p38 MAPK/miR-21 pathways in HK-2 cells	[237]
Dapagliflozin/Kidney	miR-21 miR-181a	Gentamicin-induced renal injury in Wistar rats	Dapagliflozin administration decreases markers of kidney injury, such as increased creatinine, blood urea, and urine protein	Dapagliflozin improves kidney function and structure in association with an increase in miR-21 and a decrease in miR-181a expression in kidney tissue	[242]
Empagliflozin/Kidney	miR-27a-3p, miR-190a-5p, miR-196b-5p	Murine model of early T2DM associated with obesity and insulin resistance on a high-fat diet	Empagliflozin treatment results in improved renal function parameters	Empagliflozin changes urinary exosomal miRNA profiles, including increased levels of miR-27a-3p, miR-190a-5p, miR-196b-5p	[243]
Dapagliflozin/Kidney	circRNA_012448 miR-29b-2-5p	HG-treated cultured human renal proximal tubular epithelial cells (HK-2)	The target genes of dapagliflozin-related circRNAs primarily focused on gene expression, glucose metabolic process, extracellular exosome, epigenetics, glyoxylate and dicarboxylate metabolism, amino acid metabolism, and lysosome pathways	Dapagliflozin treatment of HG-treated HK-2 cells induced changes in multiple circRNAs expression, including the hsa_circRNA_012448-hsa-miR-29b-2-5p-GSK3β pathway	[246]
Dapagliflozin/Kidney/DN	lncRNA XR_382492.4, XR_873495.3, XR-388840.1, NR-015554.2, XR-382493.3, and XR-876705.2	db/db mice model of T2DM nephropathy, RNA-seq study	Dapagliflozin improves glucose intolerance, high urinary albumin/creatinine ratio, and renal damage	Dapagliflozin treatment reversed the expression changes in XR_382492.4, XR_873495.3, XR-388840.1, NR-015554.2, XR-382493.3, and XR-876705.2 associated with DN in kidney tissue	[245]
Empagliflozin/Kidney/DN	miR-466i-3p, miR466f-3p, miR-709, miR-207, and miR-297b-3p	Murine streptozocin (STZ)–treated model of induced DN	The analysis of DEGs indicated that empagliflozin may inhibit the progression of diabetic nephropathy by inhibiting inflammation, apoptosis, and senescence	Empagliflozin treatment inhibits diabetic nephropathy according to the data of RNA-seq of kidney tissue from DN mice, empagliflozin-treated DN mice, and negative control	[239]

## 7. Genetic Variants in Non-Coding RNA in T2DM

T2DM and diabetic complications are multifactorial diseases with a strong genetic component [257]. To date, genome-wide association studies (GWAS) have identified approximately 700 variants, mainly single-nucleotide polymorphisms (SNPs), associated with predisposition to these diseases [258]. Information about SNPs associated with ncRNAs’ regulatory action in the development of T2DM also comes from sources independent of GWAS. Summarized, the largest part of already discovered SNPs affects mRNA binding sites interacting with miRNAs due to complementarity, and a smaller part pertains to substitutions in the seed sequence of the miRNAs [259].

### 7.1. T2DM-Associated Variants in miRNAs

In a study by Jahantigh et al. and colleagues, an association was found between the C rs4705342 and C rs4705343 alleles, localized in the promoter region of the *MIR143/145* microRNA gene cluster [260], with an increased and decreased risk of developing T2DM, respectively (Table 3). Moreover, individuals with the rs4705342-CC genotype have significantly higher levels of low-density lipoprotein cholesterol, fasting blood glucose, glycated hemoglobin, and miR-143 expression compared with carriers of the rs4705342-TT genotype [261]. It has previously been shown that obesity-induced increased expression of miR-143 impairs the ability of insulin to activate the phosphatidylinositol 3-kinase (PI3K)/Akt pathway in the liver and muscle tissue, causing insulin resistance and impaired glucose metabolism [262]. In addition, miR143 is involved in the process of autophagy inhibition, which can also lead to insulin resistance [263]. In turn, miR145 may play a protective role in the pathogenesis of T2DM due to a decrease in chronic inflammation. For example, in the study by He et al., miR145 expression in PBMC of T2DM patients was significantly lower than in the control group [264]. Inhibition of miR145 by intravenous administration of an antisense oligonucleotide to C57BL/6 J mice stimulated macrophage infiltration into the liver and their polarization into a proinflammatory phenotype. Restoration of miR145 expression resulted in decreased weight in mice, decreased fasting blood glucose levels, improved glucose tolerance, and reduced macrophage infiltration into the liver [264]. Thus, rs4705342 and rs4705343 are promising candidates for further study of their role in regulating *miR-143/miR-145* expression and mechanisms of T2DM development.

In the work of Sun et al., an association was established between rs8089787 and rs10877887, located in the promoter regions of *miR-133a-1* and *let-7f*, respectively, and impaired diastolic cardiac function in T2DM [265]. It has been demonstrated that let-7f mimetics can suppress the expression level of IGF1R (insulin-like growth factor 1 receptor), whereas let-7f inhibitors promote the increase in the expression level of IGF1R [266]. In addition, let-7f expression levels have been shown to be decreased in individuals with T2DM compared to healthy controls [267]. In turn, the expression level of miR-133a is higher in individuals with T2DM compared to healthy individuals. Moreover, a positive correlation was observed between the expression level of miR-133a and the levels of triglycerides, total cholesterol, fasting blood glucose, and glycated hemoglobin [268]. An association with an increased risk of T2DM has also been established with alleles of non-coding variants: rs1292037-C and rs13137-T, located in the downstream region of *miR-21* (−281 bp and −404 bp from the transcription start site, respectively) [269], and rs13283671-C in the downstream region of *miR-31* (−444 bp from the transcription start site) [270]. The rs1076064-G allele in the upstream region of *miR-378a* (+222 bp from the transcription start site) has a protective role in T2DM [270].

Some studies demonstrate discordant results. For example, a study by Khan et al. analyzed volunteers from Pakistan and found an association of the rs11614913 and rs2910164 polymorphisms located in the stem-loop miR-196a-2 (MIR196A2) and seed sequence miR-146a (MIR146A) with T2DM [271]. It was shown that the C allele of rs11614913 and the G allele of rs2910164 are associated with an increased risk of developing T2DM [271]. In contrast, other studies reported an association of the T allele of rs11614913 with an increased risk of developing cardiovascular diseases in patients with T2DM [272] and the G allele of rs2910164 with a reduced risk of developing T2DM [273]. Genetic variants that affect miRNA processing are also thought to contribute to the development of T2DM. For example, rs72631823, located in the terminal loop of pre-miR-34a, has been shown to affect the expression of the mature form of miR-34a [274]. miR-34a is involved in maintaining the functioning of pancreatic β-cells, and disruption of its activity leads to the development of endothelial dysfunction, oxidative stress, and inflammation, and predicts new-onset diabetes in coronavirus disease 2019 (COVID-19) patients [275,276]. Transfection of plasmid constructs containing the pre-miR-34a sequence with the G or A-rs72631823 allele into INS-1 (rat insulinoma) and MIN6 (mouse insulinoma) cell lines showed increased expression of miR-34a with the A allele in all cell lines compared to the G allele. This is due to the fact that the terminal loop of pre-miR-34a containing the A allele of rs72631823 is in a more relaxed form than the G allele, which leads to more efficient processing of pre-miR-34a by Drosha and Dicer ribonucleases [274].

In a study by Yang et al., it was shown that the mutant allele rs895819-C, located in the terminal loop of pre-miR-27a, leads to a decrease in the expression of both pre-miR-27a itself and its mature form in neural progenitor cells (NPCs) compared to the wild-type allele rs895819-T [277]. It is known that the C allele is associated with a reduced risk of developing T2DM [278] and coronary heart disease [279], which is one of the complications of diabetes. Given the involvement of miR-27a in the development of obesity, insulin resistance, and inflammation [280,281,282], it can be assumed that the protective role of rs895819-C in the pathogenesis of T2DM is carried out by suppressing the activity of miR-27a. It was also found that carriers of the G allele and the GG genotype of the rs3746444 polymorphism, located in the stem loop miR-499 precursor, the mature form of which is involved in the PI3K/AKT signaling pathway and glycogen synthesis, have an increased risk of developing T2DM both without and with polyneuropathy [283].

In the Han Chinese population, a 3′ UTR variant in the *TBX5* gene increased the risk for congenital heart disease, such as atrial and ventricular septal defects, by nearly two-fold [284]. The mutant allele has increased binding affinity to miR-9 and miR-30a, which reduced *TBX5* expression [284]. Circulating levels of miR-146a were significantly elevated in the cohort of 30 healthy controls compared with 30 T2DM patients [285]. The rs2910164-C allele was associated with reduced expression levels of the miR-146a but not its mRNA targets and cytokine levels in diabetic patients [285].

Novel genetic predictors of T2DM development have been increasingly discovered both by high-throughput and targeted studies. Yang et al. conducted a case–control study to evaluate the associations of *TGFBRAP1* (TGF-β1 receptor-associated protein 1) and *TGFBR2* (TGF-β1 receptor 2) SNPs with T2DM risk [286]. Aimed to study the genetic effects of SNPs on diabetes-related miRNA expression, they screened T2DM-related miRNA in the patient blood plasma by microarray, and 15 differentially expressed miRNAs were further validated in 75 T2DM and 75 normal glucose tolerance (NGT). Their findings suggest that genetic polymorphisms of TGFBRAP1, especially rs224179, may contribute to the genetic susceptibility of T2DM by mediating diabetes-related miR-30b-5p and miR-93-5p expression [286].

### 7.2. T2DM-Associated Variants in lncRNAs

Data on T2DM-associated variants in lncRNAs are still limited. Based on a number of GWAS studies in various ethnic groups, a link between rs10811661 and the risk of developing T2DM was established [287] and confirmed [288,289,290] (Table 3).

**Table 3 ijms-26-11198-t003:** NcRNA Genetics in T2DM.

SNPs/Chromosome/Involvement in SGLT2-i Effect	SNP/Gene and Its Association with Diabetes Complications	Description of the Effect	Refs
rs2076380 G>A Chromosome: 20:38165127 (GRCh38) 20:36793529 (GRCh37) Location: 20:38165127 Cytogenetic region:20q11.23 SGLT2-i –	intron *TGM2*/ lncRNA TGM2 Mapped gene(s): *TGM2* intron_variant lncRNA TGM2 is associated with T2DM No data on diabetic nephropathy and Cardiomyopathy	The AA genotype is associated with an increased risk of developing T2DM; risk allele A disrupts the secondary structure of this lncRNA, affecting its stability and the expression of TGM2 in pancreatic beta cells. Diminished LncTGM2 in human beta cells impairs glucose-stimulated insulin release.	[291]
rs7158663 A>G Chromosome: 14:100853087 (GRCh38) 14:101319424 (GRCh37) SGLT2-i –	exon *MEG3*/lnc MEG3 lncRNA MEG3 is associated with diabetic nephropathy No data on diabetic cardiomyopathy	The GG genotype is associated with a protective effect against late-stage diabetic retinopathy in patients with type 2 diabetes.	[292]
rs12427129 C>T, Chromosome: 12:53973906 (GRCh38) 12:54367690 (GRCh37) rs1899663 G>T Chromosome: 12:53967210 (GRCh38) 12:54360994 (GRCh37) SGLT2-i –	intron *HOTAIR*/lnc HOTAIR Gene: *HOXC11* *HOTAIR* lncRNA HOTAIR is associated with diabetic nephropathy and Diabetic Cardiomyopathy	The rs12427129-T allele is associated with an increased risk of diabetic retinopathy The rs1899663-TT genotype is associated with an increased risk of diabetic retinopathy	[293]
rs1333049 G>A,C Chromosome: 9:22125504 (GRCh38) 9:22125503 (GRCh37) SGLT2-i – Metformin +	found at the intron of cyclin-dependent kinase inhibitor 2B antisense RNA 1 (*CDKN2B-AS1*) lncRNA CDKN2B-AS1 is associated with diabetic nephropathy no data in diabetic cardiomyopathy	rs1333049 have shown significant association with poor metformin response rs1333049 is found to be significantly associated with HbA1c level in Mexican nonobese T2DM patients rs1333049 (SNP) in *CDKN2B-AS1*, located in the 9p21 region. is associated with advanced carotid artery atherosclerosis	[294,295,296]
rs2151280 G>A Chromosome: 9:22034720 (GRCh38) 9:22034719 (GRCh37) SGLT2-i –	intron *CDKN2B-AS1*/ lnc CDKN2B-AS1 Gene: *CDKN2B-AS1* lncRNA CDKN2B-AS1 is associated with diabetic nephropathy	The G allele is associated with an increased risk of proliferative diabetic retinopathy, decreased glomerular filtration rate, and high-density lipoprotein levels	[297]
rs2891168 A>G Chromosome: 9:22098620 (GRCh38) 9:22098619 (GRCh37) SGLT2-i –	Gene: *CDKN2B-AS1* lncRNA CDKN2B-AS1 is associated with diabetic nephropathy no data in diabetic cardiomyopathy	Susceptibility to coronary artery disease and type 2 diabetes mellitus	[298]
rs55829688 Chromosome: 1:173868168 (GRCh38) 1:173837306 (GRCh37) SGLT2-i –	Gene: *GAS5* lncRNA GAS5 is associated with diabetic nephropathy and cardiomyopathy	Association of *GAS5* gene polymorphisms with the progression of DKD. Carriers of at least one minor allele (C) of rs55829688 (TC and CC) more frequently suffer from advanced DKD than those homozygotes for the major allele (TT).	[299]
rs55829688 T>C Chromosome: 1:173868168 (GRCh38) 1:173837306 (GRCh37) SGLT2-i –	intron *GAS5*/lnc GAS5 Gene: *GAS5* lncRNA GAS5 is associated with diabetic nephropathy and cardiomyopathy	The C allele is associated with an increased risk of developing diabetic kidney disease, a decrease in the glomerular filtration rate	[299]
rs10811661 T>C Chromosome: 9:22134095 (GRCh38) 9:22134094 (GRCh37) SGLT2-i –	+13 kb from TSS *ANRIL*/lnc ANRIL lncRNA ANRIL is associated with diabetic nephropathy and cardiomyopathy	The T allele is associated with an increased risk of T2DM	[287,289,290,300,301]
		The T allele is associated with decreased insulin secretion	[302,303,304]
		The T allele is associated with increased expression of ANRIL in the islets of Langerhans	[305]
rs895819 A>G Chromosome: 19:13836478 (GRCh38) 19:13947292 (GRCh37) SGLT2-i + in Diabetic Nephropathy	miRNAs27a Gene: *MIR23AHG* *MIR23A* *MIR24-2* *MIR27A* miR-27a-3p and miR-27b are associated with T2DM, coronary artery disease and diabetic nephropathy	miR-27a rs895819-GG genotype protects from coronary artery disease CAD risk subgroup analysis demonstrated that rs895819 C allele conveyed a significant protective effect against T2DM development in Caucasians	[278,279,306]
rs3746444 Chromosome: 20:34990448 (GRCh38) 20:33578251 (GRCh37) SGLT2-i --	miR-499a Gene: *MYH7B,* *MIR499A,* *MIR499B* miR-499a is associated with T2DM and Diabetic cardiomyopathy	miR-499a rs3746444 A>G polymorphism is correlated with T2DM and diabetic polyneuropathy, with carriers of the GG genotype and the G allele being at an increased risk in the Romanian population	[283,307]
rs8089787 T>A,C,G Chromosome: 18:21826640 (GRCh38) 18:19406601 (GRCh37) SGLT2-i – rs10877887 T>C Chromosome: 12:62603400 (GRCh38) 12:62997180 (GRCh37) SGLT2-i –	miR-133a-1 Gene: *MIB1* *MIR133A1* *MIR133A1HG* miR-133a-1 is associated with myocardial fibrosis miRNA-let-7f Gene: *LINC01465* *MIRLET7I* let-7f is associated with mitochondrial dysfunction and metabolic disturbances in T2DM	miR-133a-1-rs8089787 and let-7f-rs10877887 were associated with impaired cardiac diastolic function in T2DM	[265]
rs10757278 A>G Chromosome: 9:22124478 (GRCh38) 9:22124477 (GRCh37) SGLT2-i –	intron *ANRIL*/lncRNA ANRIL lncRNA ANRIL is associated with diabetic nephropathy and diabetic cardiomyopathy	The relative fast plasma glucose (FPG) and HbA1c levels were relatively lower in individuals with the AA genotype and higher in those with the GG genotype	[308]
rs1292037 T>C Chromosome: 17:59841547 (GRCh38) 17:57918908 (GRCh37) SGLT2-i +	miR-21 Gene: *VMP1* *MIR21* miR-21 is associated with susceptibility to coronary heart disease, diabetic cardiomyopathy and diabetic nephropathy	rs1292037C allele and C/C genotype in miR-21 were strongly associated with elevated susceptibility to coronary heart disease (CHD) in a Chinese Han population	[269]
rs1292037 T>C (see above) rs13137 A>G,T Chromosome: 17:59841670 (GRCh38) 17:57919031 (GRCh37) miR-21 SGLT2-i + rs78312845 A>C,G 17:7018112 (GRCh38) 17:6921431 (GRCh37) SGLT2-i –	miR-21 Gene: *VMP1* *MIR21* miR-21 miR-21 is associated with susceptibility to diabetic cardiomyopathy and diabetic nephropathy miR-195 Gene: *MIR195* miR-195 is associated with apoptosis in Diabetic Cardiomyopathy	the C allele of rs1292037 in miR-21 could increase the risk of T2DM the T allele of rs13137 in miR-21 could be a risk factor for T2DM rs78312845 in miR-195 contributed to the level of fasting plasma glucose (FPG) and HbA1C in nondiabetic group in the Han Chinese population	[309]
rs1076064 A>G Chromosome: 5:149732603 (GRCh38) 5:149112166 (GRCh37 SGLT2-i – rs13283671 C>A,T Chromosome: 9:21511741 (GRCh38) 9:21511740 (GRCh37) SGLT2-i –	miR-378a Gene: *MIR378A* miR-378a-3p is associated with diabetic nephropathy miR-31 Gene: *MIR31* *MIR31HG* C allele could increase the risk of developing T2DM miR-31-3p is associated with diabetic nephropathy	The results showed that miR-378a rs1076064 G allele could be a protective factor in T2DM, whereas the miR-31 rs13283671 C allele could increase the risk of T2DM	[270]
rs11614913 C>T Chromosome: 12:53991815 (GRCh38) 12:54385599 (GRCh37) SGLT2-i +	exon *MIR196A2*/stem-loop miR-196a-2 Gene: *MIR196A2* miR 196b-5p is associated with diabetic nephropathy	The C allele is associated with an increased risk of T2DM	[271]
		The T allele is associated with an increased risk of cardiovascular diseases in T2DM miR196a2 C>T (rs11614913) and miR499 A>G (rs3746444) were found to be strongly associated with increased risk for CAD	[272,310]
rs72563729 G>A Chromosome: 1:1167183 (GRCh38) 1:1102563 (GRCh37) SGLT2-i +	Gene: *MIR200B* miR 200b is associated with diabetic nephropathy miRNA-200c is associated with diabetic cardiomyopathy	allelic association of *MIR200B* variations with sight-threatening diabetic retinopathy	[311]
rs2910164 C>G Chromosome: 5:160485411 (GRCh38) 5:159912418 (GRCh37) SGLT2-i – rs531564 G>C Chromosome: 8:9903189 (GRCh38) 8:9760699 (GRCh37) SGLT2-i –	exon *MIR146A*/seed sequence miR-146a Gene: *MIR146A* *MIR3142HG* miRNA-146a is associated with inflammation in diabetic cardiomyopathy Gene: *MIR124-1HG* *MIR124-1* miR-124a is a T2DM circulating marker	The G allele is associated with an increased risk of T2DM	[271]
		miR-146a rs2910164 (G allele) and miR-124a rs531564 (C allele) might function as protective factors in T2DM in Asian population	[273]
rs4705342 T>C Chromosome: 5:149428408 (GRCh38) 5:148807971 (GRCh37) rs4705343 T>C Chromosome: 5:149428518 (GRCh38) 5:148808081 (GRCh37) SGLT2-i –	promoter region of a gene cluster *MIR143*/*145*/miR143/145 Gene: *MIR143* *CARMN* upstream_transcript_variant	The C-rs4705342 and C-rs4705343 alleles are associated with an increased and decreased risk of developing T2DM, respectively The CC genotype of rs4705342 might be a risk factor in T2DM by increasing the expression of miRNA-143 in the northern Chinese Han population	[260,261]
rs72631823 G>A Chromosome: 1:9151723 (GRCh38) 1:9211782 (GRCh37) SGLT2-i – Metformin+	exon *MIR34A*/terminal loop pre-miR-34a Gene: *MIR34A* *MIR34AHG* miR-34a is associated with diabetic kidney disease	Allele A promotes the formation of a relaxed form of the pre-miR-34a terminal loop, which facilitates its processing and leads to an increase in miR-34a expression in INS-1 and MIN6 cells. The effect of allograft inflammatory factor-1 on inflammation, oxidative stress, and autophagy via miR-34a/ATG4B pathway in diabetic kidney disease	[274,312,313]
rs60432575 G>A Chromosome: 20:57895403 (GRCh38) 20:56470459 (GRCh37) SGLT2-i – Sulfonylurea drug+	*MIR4532*/hsa-miR-4532 urinary microRNA biomarkers in lupus nephritis and diabetic nephropathy	The G allele promotes efficient binding of miR-4532 to the 3′-untranslated region of KCNJ11 mRNA, resulting in decreased expression of KCNJ11 and Kir6.2 in HEK293 cells, as well as insulin secretion after stimulation of MIN6 cells with sulfonylurea drugs	[314,315,316]

rs10811661 is located 13 kb downstream of the lncRNA *ANRIL* transcription start site in a locus that includes the *MTAP*, *CDKN2A*, *CDKN2B*, and *ANRIL* genes [305]. ANRIL is known to be a negative transcriptional regulator of *CDKN2A* and *CDKN2B*, which encode inhibitors of cyclin-dependent kinases p14/p16 and p15, respectively, which are responsible for pancreatic β-cell proliferation and regeneration [317]. Mechanistically, ANRIL suppresses the transcriptional activity of these genes by recruiting the polycomb repressor complexes PRC2 and PRC1, which add the repressing histone modifications H3K27me3 and H2AK119ub [318]. Considering the availability of information on the association of rs10811661 with impaired insulin secretion and changes in ANRIL expression [305], it can be assumed that this polymorphism plays an important role in the pathogenesis of T2DM. An association with T2DM has also been shown for rs10757278, located in the *ANRIL* intron [308]. In this case–control study conducted on volunteers from China, individuals with the GG genotype were found to have a higher risk of developing T2DM. In addition, individuals with T2DM who had the GG genotype had lower HbA1C and fasting glucose levels, as well as serum ANRIL expression levels [308].

A study by Gonzalez-Moro et al. [291] showed that rs2076380, located in a DNA region corresponding to lncRNA *TGM2*, is involved in T2DM pathogenesis through allele-specific downregulation of *TGM2* gene expression in human beta cells, which impairs glucose-stimulated insulin release. The *TGM2* risk allele A, associated with T2DM, has been shown to disrupt the secondary structure of this lncRNA, affecting its stability and, consequently, TGM2 expression in pancreatic beta cells [291]. G/G genotype of lncRNA MEG3 rs7158663 single-nucleotide polymorphism is associated with a protective effect against late-stage diabetic retinopathy in T2DM patients [292]. An association has also been shown between rs12427129 and rs1899663 in lncRNA HOTAIR and the development of diabetic retinopathy [293], the rs2151280-G allele in CDKN2B-AS1 and an increased risk of proliferative diabetic retinopathy, decreased glomerular filtration rate and high-density lipoprotein levels [297], rs55829688 in lncRNA GAS5 and diabetic kidney disease [299], rs2891168 in lncRNA CDKN2B-AS1 and cardiovascular complications in diabetes [298], and rs1333049 in CDKN2B-AS1 and carotid atherosclerosis [294] and HbA1c levels [295].

Pharmacotherapy in T2D is routinely selected based on the patient’s comorbidities, medication side effects, cost, and availability, without consideration of the underlying pathophysiology driving disease phenotype [319]. Pharmacogenetics is a promising precision medicine approach to tailor treatment choices by a patient’s underlying genetic background [320,321]. Using genetics to guide medication selection might decrease the risk for comorbidities or suggest lower medication doses, reducing side effects [322]. Specific loci have been linked to the response to metformin [323,324,325], sulfonylureas [326], and GLP-1 receptor agonists [327,328,329].

Examples of SNPs in ncRNAs associated with the response to drug therapy for different diseases have been published [330,331,332]. However, the data about SNPs in ncRNAs associated with the response to antidiabetic drugs are very limited. The rs1333049-C allele, located in the *CDKN2B-AS1* intron, was found to be associated with a poor response to metformin (absolute HbA1c level changed by less than 0.5% from baseline after a three-month course of treatment) in Filipinos with T2DM [296]. Chen et al. showed that the G allele of rs60432575 located in the *MIR4532* gene promotes efficient binding of hsa-miR-4532 to the 3′-untranslated region of KCNJ11 mRNA, decreasing the expression of KCNJ11 and the Kir6.2 protein encoded by *KCNJ11* in HEK293 cells, as well as insulin secretion after stimulation of MIN6 cells with three sulfonylurea drugs (gliclazide, glibenclamide and glimepiride) [314]. In contrast, disruption of hsa-miR-4532 binding to *KCNJ11* by the A allele did not affect the expression of KCNJ11, Kir6.2, and insulin secretion [314]. Thus, further study of the role of rs60432575 in the response to sulfonylurea drugs is of great interest. According to our knowledge, there are no publications on the studies of SNPs in ncRNAs associated with the response to SGLT2-is.

### 7.3. Future Perspectives in T2DM Pharmacogenetics

As far as the genetic background of T2DM and its complications being increasingly supported, it is reasonable to expect that the genetic component will be involved in the response to SGLT2 inhibitors. Identification of the susceptible loci associated with T2DM complication risks, such as renal and cardiac comorbidities, may support early identification of T2DM patients with an increased risk, while identification of SGLT2 inhibitors targets may support targeted therapeutic interventions through the analysis of a panel of genetic markers [333]. Li et al. used canonical correlation analysis based on multivariate regression analysis on the summary statistics data from two large independent meta-analyses of GWAS [334]. This led to the identification of 4624 SNPs and 1745 genes associated with chronic kidney disease (CKD) and heart failure (HF) [334]. Validation of these genes against the transcriptome-wide gene expression data for CKD and HF was followed by gene enrichment and KEGG pathway analyses to explore their potential functional significance. This study identified 169 putative pleiotropic genes significantly associated with at least one of the two conditions. Among these genes, 21 were predicted as potential therapeutic targets of SGLT2 inhibitors in both CKD and HF [334]. It is of interest to investigate whether these genes function in the circulating monocytes or in the resident macrophages.

Transcriptome sequencing (RNA-seq) and the search for allele-asymmetric events (ASE) are currently accepted approaches to identifying SNPs with regulatory function [335,336]. Along with other approaches based on the principle of searching for allele-asymmetric events, this method allows for evaluating the functionality of the examined SNP. Moreover, it allows for reducing the number of individuals included in the study, including a single individual, since allele-asymmetric events are studied for each individual on an identical genetic background and under identical external and internal environmental conditions [335,337]. This fact makes the ASE-based approach rather attractive compared with the GWAS methodology, which is based on the genetic analysis of hundreds and thousands of patients and controls, and in most cases, pharmaceutical studies are unavailable. Therefore, an RNA-seq-based ASE search seems very promising for identifying regulatory SNPs associated with different responses to therapeutic agents. For example, a large-scale search of allele-asymmetric events for 50 different substances (hormones, nutrients, several drugs, and environmental pollutants) in transcriptome data was conducted using primary cultures of five cell types (LCLs, PBMCs, HUVECs, SMCs, and melanocytes), each represented by samples from three individuals [338]. The authors carried out an analysis of the RNA-seq data, which allowed them to identify 253 SNPs; the misbalance in the representation of the alleles in the transcriptomes either newly appeared or significantly increased in response to treatment with a particular drug. Using the same approach, 561 ASEs were identified that responded to the treatment of CD4+ lymphocytes (from 24 genotyped individuals) with immobilized antibodies to CD3/CD28 [339]. We believe that the application of this approach can significantly extend the possibilities of searching for potentially regulatory SNPs in ncRNAs and their mRNA targets, including those involved in the mechanisms of response to antidiabetic drugs. Thus, we made a bioinformatic analysis of the data from Moyerbrailean et al. [338], which demonstrated that 58 of the 253 SNPs found by the authors represent variations in the microRNA binding sites in the 3′UTR of mRNAs, which, according to calculations, significantly affect the efficiency of their binding. In particular, several discovered SNPs were potentially involved in the individual response to dexamethasone, vitamin D, and selenium-containing pharmaceuticals [332].

## 8. Conclusions

This review elucidated the state-of-the-art of candidates and accumulating data from the high-throughput studies from genetic and functional studies on the regulatory role of ncRNAs in the T2DM cardio-renal complications, and linking these data with the therapeutic effect of the SGLT2-is. Monocytes and macrophages play a central functional and regulatory role at all stages of diabetes initiation and progression, and are critically important for the development of macro- and microvascular complications. SGLT2 inhibitors can directly program the activation of monocytes and macrophages [231]. Accumulating data indicate that aberrant expression of certain ncRNAs during T2DM development under the changed metabolic conditions is associated with increased apoptosis, tissue fibrosis and hypertrophy, epithelial–mesenchymal transition, inflammation, and other pathological processes, resulting in heart and kidney injury. The expression of ncRNAs is tissue- and cell-type-specific; in this regard, various signaling pathways with their engagement have been identified for the cardiac and renal cell types. Our review highlights the need to deeply analyze the ncRNAs in macrophages, while this level of regulation has very limited attention in the macrophage field. Researchers discovered dozens of ncRNAs associated with these processes, as well as ongoing studies using whole transcriptome analysis have allowed us to add more candidate ncRNA markers and create competing RNA networks involved in T2DM complications. Since chronic inflammation is one of the important factors in T2DM pathogenesis, ncRNA levels change during the transformation of circulating monocytes, and resident macrophages in the heart and kidneys attract significant attention. Moreover, monocytes are exposed to the metabolic changes already in the circulation, and a changed ncRNA profile can serve as a first line of epigenetic programming predisposing circulating monocytes to differentiate towards the pathological macrophages; however, this mechanism is still poorly studied. The current challenge is to summarize the obtained results from different laboratories and to perform standardized mechanistic studies on the pathogenesis of T2DM cardio-renal complications at the cellular and molecular levels. This fundamental knowledge is urgently needed in order to select the proper ncRNA markers for early T2DM diagnostics and prognosis, to select novel therapeutic targets, and to identify which ncRNAs are critical for monocyte/macrophage activation under the metabolic conditions that can be controlled by SGLT2 inhibitors or can define the response of patients to SGLT2 inhibitors.

Despite the considerable interest in the action of SGLT2-is, the molecular and cellular mechanisms of their preventive effects for the development of T2DM complications are understood in a limited way, where, in particular, monocytes and macrophages deserve deep analysis. Understanding the SGLT2-i mechanisms of action, which, along with normalizing glucose levels, lead to a decrease in chronic inflammation, reducing the severity of T2DM complications, is necessary for the development of ncRNA-based therapeutic tools. Selecting not only proper ncRNAs tools or targets but also the decision-making cell type is needed to significantly improve the treatment specificity and reduce unwanted side effects. The development of ncRNA-based therapeutic approaches for T2DM is emerging, but drug prototypes have not yet reached clinical trials, although pre-clinical experimental data are promising [340]. However, summarizing the data from the publications included in our review did not allow us to extract clear mechanistic routes of the SGLT2-is’ effects in macrophages in heart and kidney diabetic complications. Further, deeper insight into the role of ncRNA and associated signaling pathways in the SGLT2-i-induced processes in the T2DM models in vivo and cellular models in vitro should close this gap and enforce clinical application of ncRNA-based therapeutics. The identification of novel genetic variants in ncRNAs and their targets associated with T2DM and its complications, with a deep analysis of ncRNA profiles and functions in monocytes and macrophages in the context of SGLT2-i treatment, are needed to develop accurate predictive algorithms for personalized therapy of T2DM patients.

## Figures and Tables

**Figure 1 ijms-26-11198-f001:**
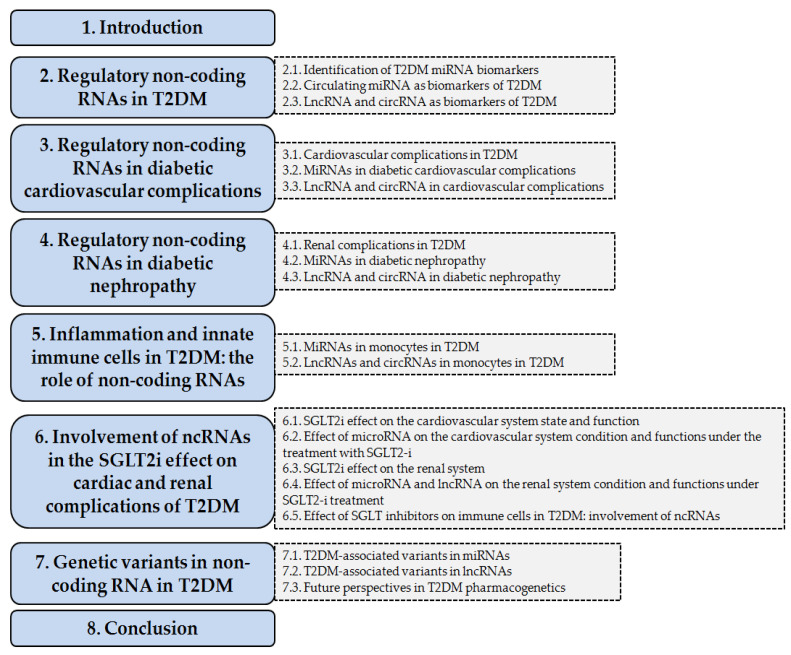
A flowchart representing the structure and content of the review.

**Figure 2 ijms-26-11198-f002:**
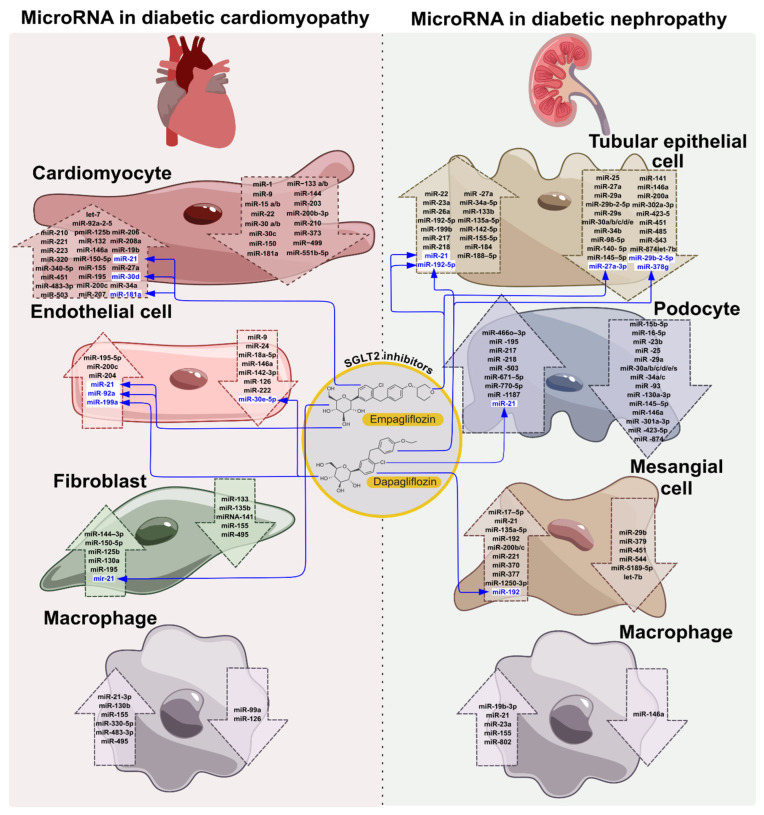
Effects of SGLT2 inhibitors via microRNA modulation in diabetic complications. The diagram illustrates the key cellular players and proposed miRNA networks in diabetic cardiomyopathy (**left**) and diabetic nephropathy (**right**). In both organs, specific cell types—including cardiomyocytes, endothelial cells, fibroblasts, podocytes, tubular epithelial cells, and mesangial cells—exhibit altered miRNA expression profiles under diabetic conditions, driving pathological processes such as fibrosis, hypertrophy, and inflammation. Infiltrating macrophages contribute to these pathways in both compartments. The model further highlights the emerging role of SGLT2 inhibitors (Empagliflozin and Dapagliflozin) in the expression and regulation of a subset of these dysregulated miRNAs (indicated by arrows), thereby mitigating cellular dysfunction and interrupting the progression of both complications.

## Data Availability

No new data were created or analyzed in this study. Data sharing is not applicable to this article.

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
