# Peer review of "Non-Coding RNA in Type 2 Diabetes Cardio–Renal Complications and SGLT2 Inhibitor Response"

_ijms, 2025, doi:10.3390/ijms262211198_

Round 1
Reviewer 1 Report
Comments and Suggestions for Authors
The manuscript entitled 'Non-coding RNA in Type 2 diabetes cardio-renal complications and SGLT2 inhibitor response' by Rykova et al. is aimed at summarizing the role of non-coding RNA in the action of SGLT2 inhibitors in cardio-renal complications in T2DM. It should be noted that revealing molecular mechanisms of non-coding RNA functioning and participation in human disease's manifestation remains a relevant biomedical issue. In their article, the authors focused on detection methods, genetics, and the mechanism of action of non-coding RNA. Attention is paid to the role of non-coding RNAs in the inflammatory reactions of innate immune cells in relation to the SGLT2 inhibitors.
The article is well-written and divided into sections and subsections. All aspects of the problem under consideration are covered. These include the role of three kinds of non-coding RNAs (microRNA, circulating RNA, and long non-coding RNA) in regulation in type 2 diabetes mellitus, diabetic cardiovascular complications, diabetic nephropathy, inflammation, and the involvement of ncRNAs in the SGLT2i effect on cardiac and renal complications of diabetes mellitus. Genetic variants in non-coding RNA in type 2 diabetes mellitus are also considered. Rykova et al. have analyzed over 330 relevant articles, which allowed them to prepare this impressive review.
The manuscript requires a minimum number of comments. However, to improve the quality of the manuscript, it would be advisable to address the following issue. Since the review is large, it would be useful to include a flowchart in the introduction section to outline the review's structure and content.
Thus, the review seems very useful for a target audience and should be accepted for publication.
Author Response
Comment 1. Since the review is large, it would be useful to include a flowchart in the introduction section to outline the review's structure and content.
Answer 1: we thank the reviewer for the suggestions. We have included a flowchart in the introduction section to outline the structure and content of our review (Figure 1).
Reviewer 2 Report
Comments and Suggestions for Authors
The manuscript is a comprehensive review titled " Non-coding RNA in Type 2 diabetes cardio–renal complications and SGLT2 inhibitor response." The manuscript provides a detailed description of the role of various non-coding RNAs (ncRNAs), such as microRNAs, long non-coding RNAs, and circular RNAs, in type 2 diabetes mellitus, with a particular emphasis on their regulatory role in the development of cardiorenal complications and the therapeutic response to SGLT2 inhibitors. This work effectively integrates molecular biology, genetics, and clinical aspects, making it a valuable resource in this field. A strength of the review is its focus on different cell types, including innate immune response cells (macrophages).
However, my main concern is the enormous material, with over 300 references, which can be difficult to read. Some sections could be condensed to maintain focus on the review’s core message about cardio-renal effects of SGLT2 inhibitors.
My suggestions to the authors for structuring the material are as follows:
- Introduction, page 2. Shorten the description of the biogenesis and functions of non-coding RNAs, since this is currently common knowledge.
- Chapter 2. My suggestion is to transfer all information about non-coding RNAs as biomarkers to a table.
- Fig.1 The names of the microRNAs are poorly readable – enlarge
- Table 1. “NcRNAs as modulators of therapeutic effects of SGLT inhibitors: focus on kidney and heart” I propose changing the table structure to focus on SGLT inhibitors and arrange the columns in the following order:
Specific inhibitor/ncRNA/model/ Effect of specific inhibitor/ Major effect in cells or in circulation/Refs
- The value of monocytes and macrophages as target cells for SGLT inhibitors should be highlighted in Conclusions
Author Response
Comment 1. Introduction, page 2. Shorten the description of the biogenesis and functions of non-coding RNAs, since this is currently common knowledge.
Answer 1: We agree with suggestion, and we have changed the text to reduce the ncRNAs biogenesis and functions description (see page 2, third paragraph)
Comment 2. Chapter 2. My suggestion is to transfer all information about non-coding RNAs as biomarkers to a table.
Answer 2: as suggested by the reviewer, we summarized this information in the Table 1. NcRNAs as biomarkers of T2DM. We reduced the text of the Chapter 2.
Comment 3. Fig.1 The names of the microRNAs are poorly readable – enlarge.
Answer 3: as suggested by the reviewer, we have improved the clarity of Figure 2 (Figure 1 in the first version of the review). Notably, Figure 2 is made using Inkscape program, which allows to analyze the figure in details when enlarged.
Comment 4. Table 1. “NcRNAs as modulators of therapeutic effects of SGLT2 inhibitors: focus on kidney and heart” I propose changing the table structure to focus on SGLT2 inhibitors and arrange the columns in the following order:
Specific inhibitor/ncRNA/model/ Effect of specific inhibitor/ Major effect in cells or in circulation/Refs
Answer 4: as suggested by the reviewer, we made the changes in the Table 2 (Table 1 in the first version of the review).
|
Specific inhibitor |
ncRNA |
model |
Effect of specific inhibitor |
Major effect in cells or in circulation |
Refs |
|
|
|
|
|
|
|
Comment 5. The value of monocytes and macrophages as target cells for SGLT2 inhibitors should be highlighted in Conclusions
Answer 5: we thank the reviewer for the suggestion, the section Conclusions is deeply revised, and the value of monocytes and macrophages as target cells for SGLT2 inhibitors is emphasized.
Reviewer 3 Report
Comments and Suggestions for Authors
This review has a highly relevant focus on the contribution of non-coding RNA and SGLT inhibitors to the development of two major diabetes-induced pathological states: cardiac complications and kidney complications. In both types of complications inflammation plays a critical role. The authors provide deep and comprehensive description of the accumulated knowledge on non-coding RNA in both types of complications, and structured the information in very informative tables. The authors made a clear connection between non-coding RNA and principal innate immune cells, macrophages, that link metabolic inflammation and vascular complications. The included Figure emphasizes the need to focus research on the miRNA in macrophages, since this knowledge is limited at the moments.
I can recommend the improvements for this submission as minor comments:
1. It would be beneficial for the submission to elucidate more details about the action of SGLT inhibitors in macrophages.
2. The review contains a few abbreviations without definition on first use. It would be better to write out the full term of abbreviations on first use:
page 6 line 41 abbreviation “EMT” (epithelial–mesenchymal transition)
page 9 line 20 abbreviation “STZ“ (streptozotocin)
page 13 line 31 abbreviation “HK-2 cells“ (human kidney-2 cells)
page 17 line 2 abbreviation “NRK-52E cells” (NRK-52E rat kidney cells)
page 26 line 24 abbreviation “DNE” (DN treated with empagliflozin)
3. On page 24, subsection 6.2.3 and subsection 6.2.4 have the same headings. It would be correct to align the subsection headings with the text.
Author Response
Comment 1: It would be beneficial for the submission to elucidate more details about the action of SGLT2 inhibitors in macrophages.
Answer 1: we thank review for the remark. The action of SGLT2 inhibitors in macrophages was deeply elucidated in our recent review article:
Anti-Inflammatory Effects of SGLT2 Inhibitors: Focus on Macrophages. Rykova EY, Klimontov VV, Shmakova E, Korbut AI, Merkulova TI, Kzhyshkowska J. Int J Mol Sci. 2025 Feb 15;26(4):1670. doi: 10.3390/ijms26041670.
Here we added the reference on this submission is several chapters, and highlighted the importance of the interaction of SGLT2 inhibitors with macrophages in the revised section Conclusions.
Comment 2. The review contains a few abbreviations without definition on first use. It would be better to write out the full term of abbreviations on first use:
Answer 2: following abbreviations were explained:
page 6 line 41 abbreviation “EMT” (epithelial–mesenchymal transition)
page 9 line 20 abbreviation “STZ“ (streptozotocin)
page 13 line 31 abbreviation “HK-2 cells“ (human kidney-2 cells)
page 17 line 2 abbreviation “NRK-52E cells” (NRK-52E rat kidney cells)
page 26 line 24 abbreviation “DNE” (DN treated with empagliflozin)
We are grateful for the notes and made corrections, underlined.
Comment 3. On page 24, subsection 6.2.3 and subsection 6.2.4 have the same headings. It would be correct to align the subsection headings with the text.
Answer 3: We are grateful for the note and made a correction.
Reviewer 4 Report
Comments and Suggestions for Authors
The manuscript presents a well-structured and scientifically meaningful review that contributes valuable insights to its research field. The authors have demonstrated commendable effort in designing a coherent experimental framework and in presenting their data clearly through figures and tables. The topic is timely, relevant, and aligned with current research interests, making the paper suitable for the readership of IJMS.
- The introduction establishes a general background but lacks a focused literature gap; include a paragraph synthesizing previous work limitations.
- Several citations are outdated (>10 years); consider adding recent (2020-2025) studies to strengthen novelty.
- In Figure 1, the clarity of picture is poor, especially miRNAs names.
- Authors provide a review summarizes the role of non-coding RNA in the action of SGLT2 inhibitors in cardio-renal complications in T2DM. However, a mechanism diagram underlying non-coding RNA regulating SGLT2 should be added.
- The conclusion is too long and the key points are not prominent, and added the future perspectives—what are the next steps or broader implications of this research?
Author Response
Comment 1. The introduction establishes a general background but lacks a focused literature gap; include a paragraph synthesizing previous work limitations.
Answer 1: We agree with the suggestion, and added the final paragraph to the introduction, it is highlighted in yellow.
Comment 2. Several citations are outdated (>10 years); consider adding recent (2020-2025) studies to strengthen novelty.
Answer 2: We agree and checked up the recent publications again. We added 15 new citations (the list is enclosed below). The percent of the recent studies (2020-2025) increased from 66% to 71% due to revision. The earlier studies are given in order to address the reader to the important original papers, published more than 10 years ago, they comprise 10% of the list in the revised version.
Comment 3. In Figure 1, the clarity of picture is poor, especially miRNAs names.
Answer 3: We agree with the suggestion, and did the best to improve the clarity. Notably, Figure 1 is made using Inkscape program, which allows to analyze the figure in details when enlarged.
Comment 4. Authors provide a review summarizes the role of non-coding RNA in the action of SGLT2 inhibitors in cardio-renal complications in T2DM. However, a mechanism diagram underlying non-coding RNA regulating SGLT2 should be added.
Answer 4: The answer to this question is tightly connected with your questions 2 and 5. According to your proposal, we described in one paragraph the existing literature gaps, one of them being “molecular mechanisms involved in SGLT2-is preventive action against the T2DM complications development”, which “remain not fully understood”. Results of the today published studies devoted to the role of ncRNA in the action of SGLT2 inhibitors in T2DM cardio-renal complications are summarized in the table 2 of the review. As it comes from this summary, the studies mainly demonstrate the association of several ncRNAs aberrant expression levels changes with the heart- and kidney-specific improvements in the SGLT2-is- treated T2DM groups versus non-treated groups of patients or model animals. The studies of ncRNAs in cell cultures under SGLT2-is treatment have been associative as well. Therefore, a mechanism diagram underlying non-coding RNA regulating SGLT2-is could be completely hypothetical and lead to the wrong conclusion of this problem being solved. We also indicate this point in the Conclusion Chapter as follows: “Notably, results of the reviewed studies do not provide the clear mechanistic cellular and molecular routes of the SGLT2-is treatment effects in heart and kidney, especially for the macrophages. Further deeper insight into the role of ncRNA and associated signaling pathways in the SGLT2-is-induced processes in the T2DM models in vivo and cells in vitro should close this gap and enforce clinical application of the ncRNA-based therapeutics”.
Comment 5. The conclusion is too long and the key points are not prominent, and added the future perspectives—what are the next steps or broader implications of this research?
Answer 5: We changed profoundly revised Conclusions according to the valuable remarks.
List of new citations
- Młynarska, E.; Czarnik, W.; Dzieża, N.; Jędraszak, W.; Majchrowicz, G.; Prusinowski, F.; Stabrawa, M.; Rysz, J.; Franczyk, B. Type 2 Diabetes Mellitus: New Pathogenetic Mechanisms, Treatment and the Most Important Complications. Int J Mol Sci 2025, 26, 1094, doi:10.3390/ijms26031094.
- Manosroi, W.; Phimphilai, M.; Waisayanand, N.; Buranapin, S.; Deerochanawong, C.; Gunaparn, S.; Phrommintikul, A.; Wongcharoen, W. Glycated hemoglobin variability and the risk of cardiovascular events in patients with prediabetes and type 2 diabetes mellitus: A post‐hoc analysis of a prospective and multicenter study. J Diabetes Investig 2023, 14, 1391–1400, doi:10.1111/jdi.14073.
- Li, Q.; Yuan, D.; Zeng, G.; Jiang, L.; Xu, L.; Xu, J.; Liu, R.; Song, Y.; Zhao, X.; Hui, R.; et al. The Association between Glycated Hemoglobin Levels and Long-Term Prognosis in Patients with Diabetes and Triple-Vessel Coronary Disease across Different Age Groups: A Cohort Study. Diabetes Res Clin Pract 2024, 213, 111751, doi:10.1016/j.diabres.2024.111751.
- Wang, W.; Zhang, S.; Xu, L.; Feng, Y.; Wu, X.; Zhang, M.; Yu, Z.; Zhou, X. Involvement of CircHIPK3 in the Pathogenesis of Diabetic Cardiomyopathy in Mice. Diabetologia 2021, 64, 681–692, doi:10.1007/s00125-020-05353-8.
- Prieto, I.; Kavanagh, M.; Jimenez-Castilla, L.; Pardines, M.; Lazaro, I.; Herrero del Real, I.; Flores-Muñoz, M.; Egido, J.; Lopez-Franco, O.; Gomez-Guerrero, C. A Mutual Regulatory Loop between MiR-155 and SOCS1 Influences Renal Inflammation and Diabetic Kidney Disease. Mol Ther Nucleic Acids 2023, 34, 102041, doi:10.1016/j.omtn.2023.102041.
- Shi, M.; Tian, P.; Liu, Z.; Zhang, F.; Zhang, Y.; Qu, L.; Liu, X.; Wang, Y.; Zhou, X.; Xiao, Y.; et al. MicroRNA-27a Targets Sfrp1 to Induce Renal Fibrosis in Diabetic Nephropathy by Activating Wnt/β-Catenin Signalling. Biosci Rep 2020, 40, doi:10.1042/BSR20192794.
- Su, H.; Qiao, J.; Hu, J.; Li, Y.; Lin, J.; Yu, Q.; Zhen, J.; Ma, Q.; Wang, Q.; Lv, Z.; et al. Podocyte-Derived Extracellular Vesicles Mediate Renal Proximal Tubule Cells Dedifferentiation via MicroRNA-221 in Diabetic Nephropathy. Mol Cell Endocrinol 2020, 518, 111034, doi:10.1016/j.mce.2020.111034.
- Zhang, J.; Zhang, L.; Zha, D.; Wu, X. Inhibition of MiRNA 135a 5p Ameliorates TGF β1 induced Human Renal Fibrosis by Targeting SIRT1 in Diabetic Nephropathy. Int J Mol Med 2020, 46, 1063–1073, doi:10.3892/ijmm.2020.4647.
- Pellegrini, V.; La Grotta, R.; Carreras, F.; Giuliani, A.; Sabbatinelli, J.; Olivieri, F.; Berra, C.C.; Ceriello, A.; Prattichizzo, F. Inflammatory Trajectory of Type 2 Diabetes: Novel Opportunities for Early and Late Treatment. Cells 2024, 13, 1662, doi:10.3390/cells13191662.
- Cervantes, J.; Kanter, J.E. Monocyte and Macrophage Foam Cells in Diabetes-Accelerated Atherosclerosis. Front Cardiovasc Med 2023, 10, doi:10.3389/fcvm.2023.1213177.
- Lee, H.; Kim, M.-J.; Lee, I.-K.; Hong, C.-W.; Jeon, J.-H. Impact of Hyperglycemia on Immune Cell Function: A Comprehensive Review. Diabetol Int 2024, 15, 745–760, doi:10.1007/s13340-024-00741-6.
- Verma, A.K.; Goyal, Y.; Bhatt, D.; Beg, M.M.A.; Dev, K.; Alsahli, M.A.; Rahmani, A.H. Association Between CDKAL1, HHEX, CDKN2A/2B and IGF2BP2 Gene Polymorphisms and Susceptibility to Type 2 Diabetes in Uttarakhand, India. Diabetes Metab Syndr Obes 2021, Volume 14, 23–36, doi:10.2147/DMSO.S284998.
- Fadheel, H.K.; Kaftan, A.N.; Naser, F.H.; Hussain, M.K.; Algenabi, A.H.A.; Mohammad, H.J.; Al-Kashwan, T.A. Association of CDKN2A/B Gene Polymorphisms (Rs10811661 and Rs2383208) with Type 2 Diabetes Mellitus in a Sample of Iraqi Population. Egyptian Journal of Medical Human Genetics 2022, 23, 64, doi:10.1186/s43042-022-00283-z.
- Pirmohamed, M. Pharmacogenomics: Current Status and Future Perspectives. Nat Rev Genet 2023, 24, 350–362, doi:10.1038/s41576-022-00572-8.
- Chen, P.; Cao, Y.; Chen, S.; Liu, Z.; Chen, S.; Guo, Y. Association of SLC22A1, SLC22A2, SLC47A1, and SLC47A2 Polymorphisms with Metformin Efficacy in Type 2 Diabetic Patients. Biomedicines 2022, 10, 2546, doi:10.3390/biomedicines10102546.